# Revisiting Tree-Sliced Wasserstein Distance Through the Lens of the Fermat–Weber Problem

**Viet-Hoang Tran**[*]
Department of Mathematics
National University of Singapore
hoang.tranviet@u.nus.edu

**Thanh Q. Tran**[*]
Department of Computer Science
National University of Singapore
thanhtran@u.nus.edu

**Thanh Chu**[*]
Department of Computer Science
National University of Singapore
thanh.chu@u.nus.edu

**Trung-Khang Tran**
Department of Computer Science
National University of Singapore
TTK0106@u.nus.edu

**Duy-Tung Pham**
FPT Software AI Center
Vietnam
tungpd10@fpt.com

**Tam Le**[†]
Department of Advanced Data Science
Institute of Statistical Mathematics
tam@ism.ac.jp

**Tan Minh Nguyen**[†]
Department of Mathematics
National University of Singapore
tanmn@nus.edu.sg

## Abstract

Tree-Sliced methods have emerged as an efficient and expressive alternative to the traditional Sliced Wasserstein distance, replacing one-dimensional projections with tree-structured metric spaces and leveraging a splitting mechanism to better capture the underlying topological structure of integration domains while maintaining low computational cost. At the core of this framework is the Tree-Sliced Wasserstein (TSW) distance, defined over probability measures in Euclidean spaces, along with several variants designed to enhance its performance. A fundamental distinction between SW and TSW lies in their sampling strategies– a component explored in the context of SW but often overlooked in comparisons. This omission is significant: whereas SW relies exclusively on directional projections, TSW incorporates both directional and positional information through its tree-based construction. This enhanced spatial sensitivity enables TSW to reflect the geometric structure of the underlying data more accurately. Building on this insight, we propose a novel variant of TSW that explicitly leverages positional information in its design. Inspired by the classical Fermat–Weber problem– which seeks a point minimizing the sum of distances to a given set of points–we introduce the Fermat–Weber Tree-Sliced Wasserstein (FW-TSW) distance. By incorporating geometric median principles into the tree construction process, FW-TSW notably further improves the performance of TSW while preserving its low computational cost. These improvements are empirically validated across diverse experiments, including diffusion model training and gradient flow. Our code is available at https://github.com/thanhquangtran/FW-TSW.

---

[*]Equal contribution
[†]Co–last author. Please correspond to: hoang.tranviet@u.nus.edu and tanmn@nus.edu.sg

# 1 INTRODUCTION

Optimal Transport (OT) (Villani, 2008; Peyré et al., 2019) has established itself as a foundational framework for comparing probability measures in a way that respects the underlying geometry of the data. By extending ground cost metrics from supports to entire distributions, OT has enabled a wide range of applications across machine learning (Bunne et al., 2022; Fan et al., 2022), data valuation (Just et al., 2023; Kessler et al., 2025), multimodal data analysis (Park et al., 2024; Luong et al., 2024), statistics (Mena & Niles-Weed, 2019; Weed & Berthet, 2019; Wang et al., 2022; Liu et al., 2022; Nguyen et al., 2022; Nietert et al., 2022), and computer vision and graphics (Lavenant et al., 2018; Saleh et al., 2022; Solomon et al., 2015). Despite its theoretical elegance and flexibility, a major practical limitation of OT lies in its computational complexity, which grows supercubically with the number of support points (Peyré et al., 2019).

To address this issue, the Sliced Wasserstein (SW) distance (Rabin et al., 2011; Bonneel et al., 2015) has been proposed as a scalable alternative. SW reduces computational cost by projecting high-dimensional probability measures onto one-dimensional subspaces, where closed-form solutions to the OT problem are available. This projection-based strategy leads to significant computational gains and has inspired a large body of research aimed at refining various components of the SW framework. Advances include accelerated sampling strategies (Nadjahi et al., 2021; Nguyen et al., 2024a; 2020), projection direction selection (Deshpande et al., 2019), and extensions to generalized integration domains (Kuchment, 2006; Kolouri et al., 2019; Chen et al., 2022; Bonet et al., 2023).

However, the restriction to one-dimensional projections may fail to capture complex geometric or topological features of high-dimensional distributions. In response, recent work has proposed using more expressive integration domains for OT, including Euclidean subspaces (Alvarez-Melis et al., 2018; Paty & Cuturi, 2019; Niles-Weed & Rigollet, 2022), tree metric spaces (Le et al., 2019; Le & Nguyen, 2021; Tran et al., 2024e), graphs (Le et al., 2022), spheres (Quellmalz et al., 2023; Bonet et al., 2022; Tran et al., 2024c), and hyperbolic spaces (Bonet et al., 2023). Among these approaches, the Tree-Sliced Wasserstein (TSW) framework (Tran et al., 2024e; 2025a) replaces directional projections with tree systems. These systems provide a structured integration domain that captures both spatial and directional information. By leveraging efficient OT solvers on tree metric spaces (Indyk & Thaper, 2003; Le et al., 2019; Le & Nguyen, 2021), TSW retains the computational advantages of SW while offering greater geometric expressiveness.

A key distinction between SW and TSW lies in their sampling strategies. While this aspect has been well studied for SW, it is often overlooked in comparative analyses. The difference is significant: SW relies solely on directional projections, whereas TSW incorporates both direction and position through its tree-based construction. This added spatial structure enables TSW to more effectively capture the geometry of data distributions. Consequently, the quality of the sampling strategy is crucial to realizing the full potential of TSW.

Standard implementations of SW (Bonneel et al., 2015) rely on uniform sampling over the hypersphere. However, this strategy does not distinguish between informative and uninformative directions (Deshpande et al., 2019; Nguyen et al., 2024b; Tran et al., 2024d; Nguyen & Ho, 2024), which may limit its practical effectiveness. To enhance performance, several studies have proposed data-informed slicing distributions, including both fixed (Nguyen et al., 2024b; Tran et al., 2024d; Nguyen & Ho, 2024) and trainable (Deshpande et al., 2019; Nguyen et al., 2020) variants. While trainable approaches yield empirical gains, they often rely on iterative optimization, which is computationally costly and may exhibit instability (Nguyen et al., 2020).

Incorporating similar sampling enhancements into TSW poses additional challenges. Unlike SW, the sampling space in TSW involves both directional and positional components, the latter corresponding to the intersection point of the tree system. As $\mathbb{R}^d$ is non-compact, there is no canonical uniform distribution analogous to that on $\mathbb{S}^{d-1}$. This complicates the design of efficient and principled sampling strategies for TSW. Existing TSW variants (Tran et al., 2024e; 2025a;d) rely on heuristic sampling schemes, which may not fully exploit the positional information encoded in the tree structure.

**Contributions.** Building on this insight, we propose a novel variant of the TSW framework that explicitly incorporates positional information into its slice distribution. Our approach is motivated by a classical problem in location theory—the Fermat–Weber problem—and aims to improve upon

existing heuristic methods by aligning the sampling distribution with the geometric structure of the data. The paper is organized as follows:

1. In Section 2, we recall the concepts of the SW and TSW distances, both of which serve as computationally efficient alternatives to the classical Wasserstein distance.

2. In Section 3, we examine the sampling strategies used in the SW and TSW frameworks. We emphasize the role of positional information in TSW, in contrast to the purely directional sampling in SW. Furthermore, we revisit the Fermat–Weber problem and the concept of the geometric median, and explain how these ideas inform the design of improved sampling distributions for tree systems in TSW.

3. In Section 4, we formally introduce the Fermat–Weber Tree-Sliced Wasserstein (FW-TSW) framework. We analyze its theoretical properties and computational complexity.

4. In Section 5, we illustrate advantages of the proposed approach on gradient flow and diffusion models, and conclude our work in Section 6. The results highlight its practical effectiveness and computational efficiency across both image-based and distributional learning scenarios.

The Appendix contains all supplementary materials, including theoretical background, detailed proofs, experimental setups with extended tables and figures, as well as a table of notation.

## 2 SAMPLING PROCESSES IN SLICED AND TREE-SLICED WASSERSTEIN DISTANCES

Let $d$ denote the dimension. Consider two probability measures $\mu$ and $\nu$ on $\mathbb{R}^d$ with corresponding density functions $f_\mu$ and $f_\nu$. We review the main ideas behind the Sliced Wasserstein and Tree-Sliced Wasserstein distances, which provide efficient alternatives to the classical Wasserstein distance.

### 2.1 REVIEW ON SLICED WASSERSTEIN DISTANCE

**Motivation.** A line in $\mathbb{R}^d$ is uniquely determined by a direction $\theta \in \mathbb{S}^{d-1}$ and a point $x \in \mathbb{R}^d$ through which it passes. Importantly, the OT problem between two probability measures supported on one-dimensional lines admits a closed-form solution. Leveraging this property, the SW framework projects high-dimensional measures onto one-dimensional lines, computes the Wasserstein distance in each projected space, and aggregates the results via averaging (Rabin et al., 2011; Bonneel et al., 2015). Since the projection depends only on the direction of the line, it suffices to consider projections parametrized by directions in $\mathbb{S}^{d-1}$.

**Radon Transform.** Consider a function $f \in L^1(\mathbb{R}^d)$. For direction $\theta \in \mathbb{S}^{d-1}$, define the function

$$\mathcal{R}_\theta f \colon \mathbb{R} \longrightarrow \mathbb{R}, \quad \mathcal{R}_\theta f(t) = \int_{\mathbb{R}^d} f(x) \cdot \delta(t - \langle x, \theta \rangle)\, dx, \tag{1}$$

where $\delta$ denotes the Dirac delta distribution. The full Radon transform is the operator

$$\mathcal{R} \colon L^1(\mathbb{R}^d) \longrightarrow \bigsqcup_{\theta \in \mathbb{S}^{d-1}} L^1(\mathbb{R}), \quad f \longmapsto \mathcal{R}_\theta f, \tag{2}$$

This construction provides a formal mechanism for projecting measures onto one-dimensional lines.

**Sliced Wasserstein Distance.** For $p \geq 1$, the Sliced $p$-Wasserstein distance (Bonneel et al., 2015) (SW$_p$) between $\mu$ and $\nu$ is defined as

$$\mathrm{SW}_p(\mu, \nu) = \left( \int_{\mathbb{S}^{d-1}} \mathrm{W}_p^p \left( \mathcal{R}_\theta f_\mu, \mathcal{R}_\theta f_\nu \right)\, d\sigma(\theta) \right)^{\frac{1}{p}}, \tag{3}$$

where $\sigma = \mathcal{U}(\mathbb{S}^{d-1})$ denotes the uniform probability measure on the unit sphere $\mathbb{S}^{d-1}$.

## 2.2 Review on Tree-Sliced Wasserstein Distance

We adopt the formulation of the Tree-Sliced Wasserstein distance introduced in Tran et al. (2024e; 2025a).[1] For a complete description, we refer the reader to Appendix A.

**Motivation.** The OT problem between two probability measures supported on a tree metric space (Semple & Steel, 2003; Le et al., 2019) admits a closed-form solution, similar to the one-dimensional case used in the SW framework. However, identifying suitable tree metric structures in $\mathbb{R}^d$ that permit efficient computation, analogous to projecting along directions $\theta \in \mathbb{S}^{d-1}$ in SW, is nontrivial. To address this challenge, Tran et al. (2024e; 2025a) introduced a class of structures known as *tree systems*, which enable efficient computation of OT on tree metrics. Informally, a tree system is a collection of $k$ one-dimensional lines in $\mathbb{R}^d$ arranged with a fixed tree topology. For simplicity, we may, for now, regard a tree system as an element of $(\mathbb{R}^d \times \mathbb{S}^{d-1})^k$, that is, a collection of $k$ lines, without explicitly considering the underlying tree structure. We denote a tree system by $\mathcal{T}$ and the set of all such $k$-line tree systems by $\mathbb{T}$. Leveraging this structure, the TSW framework projects high-dimensional probability measures onto the lines of a given tree system, solves the induced OT problem on this tree system, and aggregates the results, analogous to the averaging process in the SW framework.

**Radon Transform on Tree Systems.** Define $\mathcal{C}(\mathbb{R}^d \times \mathbb{T}, \Delta_{k-1})$ as the set of continuous maps from $\mathbb{R}^d \times \mathbb{T}$ to the $(k-1)$-dimensional standard simplex $\Delta_{k-1}$, referred to as *splitting maps*. We fix a splitting map, denoted by $\alpha$. Consider a function $f \in L^1(\mathbb{R}^d)$. For each $\mathcal{T} \in \mathbb{T}$, define the function

$$\mathcal{R}^\alpha_\mathcal{T} f \colon \bigsqcup_{i=1}^k \mathbb{R} \longrightarrow \mathbb{R}, \qquad \mathcal{R}^\alpha_\mathcal{T} f(t_i) = \int_{\mathbb{R}^d} f(y) \cdot \alpha(y, \mathcal{T})_i \cdot \delta\left(t_i - \langle y - x_i, \theta_i \rangle\right) \, dy. \quad (4)$$

The Radon Transform on Tree Systems is the operator

$$\mathcal{R}^\alpha \colon L^1(\mathbb{R}^d) \longrightarrow \prod_{\mathcal{T} \in \mathbb{T}} L^1(\mathcal{T}), \qquad f \mapsto (\mathcal{R}^\alpha_\mathcal{T} f)_{\mathcal{T} \in \mathbb{T}}, \quad (5)$$

This construction provides a formal mechanism for projecting measures onto tree systems.

**Tree-Sliced Wasserstein Distance.** The Tree-Sliced Wasserstein distance between $\mu, \nu$ is defined by

$$\mathrm{TSW}(\mu, \nu) = \int_\mathbb{T} \mathrm{W}_1\left(\mathcal{R}^\alpha_\mathcal{T} f_\mu, \mathcal{R}^\alpha_\mathcal{T} f_\nu\right) \, d\sigma_\mathbb{T}(\mathcal{T}). \quad (6)$$

Here, $\sigma_\mathbb{T}$ denotes a probability distribution over the space of tree systems $\mathbb{T}$. The construction of both the splitting map $\alpha$ and the distribution $\sigma_\mathbb{T}$ is detailed in Section 3. For clarity, we present the formulation for the case $p = 1$. When $p > 1$, the Wasserstein distance $\mathrm{W}_p(\mu, \nu)$ generally does not admit a closed-form solution as in the $p = 1$ case. Efforts to derive such expressions for $p > 1$ have led to the development of Sobolev Transport (Le et al., 2022) (ST), which differs from $\mathrm{W}_p$. While ST remains a valid metric over the space $\mathcal{P}(\mathcal{T})$, we restrict our attention to the case $p = 1$ in this work, as the generalization to higher $p$ values follows analogously.

## 3 Sampling Trees Through the Lens of the Fermat-Weber Problem

We review the sampling processes in both the SW and TSW frameworks, highlighting TSW's positional dependence in contrast to SW. We then revisit the Fermat–Weber problem and the geometric median, and discuss how this notion can guide the sampling of tree systems in TSW.

---

[1]For brevity, we refer collectively to the formulations in Tran et al. (2024e; 2025a) as the Tree-Sliced Wasserstein (TSW) distance. This terminology departs from the original notion introduced in Le et al. (2019); Le & Nguyen (2021); Yamada et al. (2022); Sato et al. (2020); Takezawa et al. (2022); Indyk & Thaper (2003); Lin et al. (2025), which was primarily developed for static-support measures in settings such as classification and topological data analysis. In contrast, TSW-SL (Tran et al., 2024e) and Db-TSW (Tran et al., 2025a) are formulated as optimal transport problems over tree systems, specifically designed to handle dynamic-support measures, as commonly found in generative modeling tasks.

### 3.1 Monte Carlo Approximation of SW and TSW Distances

**Sampling Slices in SW.** To approximate the intractable integral in Equation (3) of SW, Monte Carlo method is used as follows:

$$\widehat{\mathrm{SW}}_p(\mu, \nu) = \left( \frac{1}{L} \sum_{l=1}^{L} \mathrm{W}_p^p(\mathcal{R}_{\theta_l} f_\mu, \mathcal{R}_{\theta_l} f_\nu) \right)^{\frac{1}{p}}, \tag{7}$$

where $\theta_1, \ldots, \theta_L$ are drawn independently from $\sigma$. Since the hypersphere $\mathbb{S}^{d-1}$ is compact, $\sigma$ is commonly chosen to be the uniform distribution on $\mathbb{S}^{d-1}$ (Bonneel et al., 2015). While this choice is convenient due to its ease of sampling, it fails to differentiate between informative and uninformative projection directions when comparing probability measures (Deshpande et al., 2019; Nguyen et al., 2024b; Tran et al., 2024d; Nguyen & Ho, 2024). The conventional SW distance thus relies on a flat prior over directions, which can limit its discriminative power. To address this, alternative formulations propose selecting $\sigma$ from a parametric family of distributions over $\mathbb{S}^{d-1}$, aiming to maximize the expected sliced distance (Nguyen et al., 2020). Although such data-adaptive slicing distributions can enhance performance, identifying the optimal $\sigma$ typically involves iterative procedures that are computationally intensive and may exhibit instability.

**Sampling Slices in TSW.** As with the SW framework, the TSW distance can be approximated by randomly sampling $L$ tree systems $\mathcal{T}_1, \ldots, \mathcal{T}_L$ independently from the distribution $\sigma_{\mathbb{T}}$. In this case, the integral in Equation (6) is approximated as:

$$\widehat{\mathrm{TSW}}(\mu, \nu) = \frac{1}{L} \sum_{l=1}^{L} \mathrm{W}_1 \left( \mathcal{R}_{\mathcal{T}_l}^\alpha f_\mu, \mathcal{R}_{\mathcal{T}_l}^\alpha f_\nu \right). \tag{8}$$

We first describe the specific tree structure employed in the TSW framework. Tran et al. (2024e) proposed a general and inductive procedure for sampling arbitrary tree topologies. Building on this, Tran et al. (2025a) introduced a simplified construction that retains the representational power of TSW while allowing for efficient implementation. In this formulation, each tree system consists of $k$ lines intersecting at a common point. Accordingly, a tree system can be represented as a tuple $\mathcal{T} = (x, \theta_1, \theta_2, \ldots, \theta_k) \in \mathbb{R}^d \times (\mathbb{S}^{d-1})^k$, where $x \in \mathbb{R}^d$ denotes the intersection point (or root), and each $\theta_i \in \mathbb{S}^{d-1}$ specifies the direction of the $i^{\text{th}}$ line passing through $x$. Figure 1 (left) illustrates these tree structures.

The slicing distribution $\sigma$ over tree systems is modeled as a product of $k + 1$ independent components: one distribution over $\mathbb{R}^d$ for sampling the root point, and $k$ independent distributions over $\mathbb{S}^{d-1}$ for sampling line directions. While the directional components can be chosen as uniform over $\mathbb{S}^{d-1}$, the non-compactness of $\mathbb{R}^d$ precludes a uniform distribution. To address this, Tran et al. (2024e; 2025a) propose sampling the intersection point from a Gaussian centered at the data mean, which helps prevent the projection of nearby points to distant locations in the tree system.

However, this sampling strategy inherently constrains the ability to capture positional variability—a key advantage of TSW over SW. In addition, the splitting maps commonly employed in Tran et al. (2025a) are explicitly position-dependent. For instance, the splitting map $\alpha$ used in Tran et al. (2025a) is defined as

$$\alpha(y, \mathcal{T}) = \mathrm{softmax} \left( \{\xi \cdot d(y, \mathcal{T})_i\}_{i=1,\ldots,k} \right), \tag{9}$$

where $d(y, \mathcal{T})_i$ denotes the Euclidean distance from the point $y \in \mathbb{R}^d$ to the $i^{\text{th}}$ line in the tree system $\mathcal{T}$, and $\xi \in \mathbb{R}$ is a tunable parameter. Intuitively, under the Radon transform defined on a tree system, the mass at $y$ is distributed among its projections onto the $k$ lines in $\mathcal{T}$, weighted proportionally (or inversely proportionally) to their distances, depending on the sign of $\xi$.

### 3.2 The Fermat-Weber Problem

The preceding discussion underscores the critical role of sampling in the TSW framework, particularly when positional information is incorporated. Since TSW aims to align a source distribution with a target data distribution, it is desirable for the intersection points $x$ to minimize their average distance to the data. This naturally leads to the classical *Fermat–Weber problem* in location theory, which seeks a point that minimizes the weighted sum of distances to a set of target points.

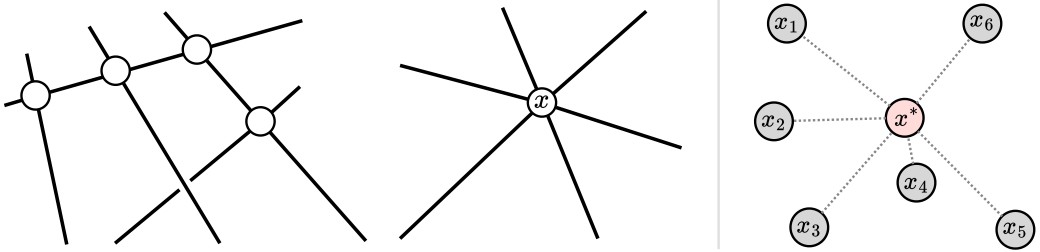

Figure 1: *(Left)* Illustration of tree structures used in Tran et al. (2024e) and Tran et al. (2025a). The structure in Tran et al. (2024e) is more general, as the framework is applicable to arbitrary tree topologies. In contrast, Tran et al. (2025a) focuses on trees formed by a set of concurrent lines, despite the fact that the underlying framework supports general trees, due to implementation considerations. *(Right)* Illustration of the geometric median. Given six points $x_i, i = 1, \dots, 6$ (shown in gray), the geometric median $x^*$ (shown in pink) is the point that minimizes the sum of distances to all six points, i.e., the total length of the connecting segments.

**The Fermat–Weber Problem.** Given a probability measure $\lambda$ on $\mathbb{R}^d$, the Fermat–Weber problem is defined as the following optimization problem:

$$x^* = \operatorname*{argmin}_{x \in \mathbb{R}^d} \int_{\mathbb{R}^d} \|x - y\|_2 \, d\lambda(y). \tag{10}$$

When $\lambda$ is approximated via Monte Carlo sampling from $n$ data points $\{x_i\}_{i=1}^n$, the problem reduces to the discrete form:

$$x^* = \operatorname*{argmin}_{x \in \mathbb{R}^d} \frac{1}{n} \sum_{i=1}^n \|x - x_i\|_2. \tag{11}$$

Such an optimal point is referred to as the *geometric median*. An illustration of this concept is provided in Figure 1 (right).

**Weiszfeld's Algorithm.** The Weiszfeld algorithm provides an efficient iterative method for approximating the geometric median. Starting from an initial estimate $x^{(0)} \in \mathbb{R}^d$, define:

$$x^{(t+1)} = \left( \sum_{i=1}^n \frac{x_i}{\|x^{(t)} - x_i\|_2} \right) \Big/ \left( \sum_{i=1}^n \frac{1}{\|x^{(t)} - x_i\|_2} \right), \quad t = 0, 1, 2, \dots, \tag{12}$$

The iteration continues until convergence, typically measured by the stopping criterion:

$$\|x^{(t+1)} - x^{(t)}\|_2 \le \varepsilon, \tag{13}$$

for some pre-specified threshold $\varepsilon > 0$. The complete procedure is summarized in Algorithm 1.

---

**Algorithm 1** Weiszfeld's Algorithm for Geometric Median

---

1: **Input:** Data points $\{x_1, \dots, x_n\}$ in $\mathbb{R}^d$, initial estimate $x^{(0)} \in \mathbb{R}^d$, tolerance $\varepsilon > 0$
2: **Output:** Approximate geometric median $x^*$
3: Set iteration counter $t \leftarrow 0$
4: **repeat**
5:     Compute update: $x^{(t+1)} \leftarrow \left( \sum_{i=1}^n \frac{x_i}{\|x^{(t)} - x_i\|_2} \right) \Big/ \left( \sum_{i=1}^n \frac{1}{\|x^{(t)} - x_i\|_2} \right)$
6:     Increment $t \leftarrow t + 1$
7: **until** $\|x^{(t)} - x^{(t-1)}\|_2 \le \varepsilon$
8: **return** $x^* = x^{(t)}$

---

The general formulation for the Fermat-Weber problem is presented in Appendix B.

**Application to the TSW Framework.** We now define a distribution over the space of tree systems $\mathbb{T}$ using the geometric median. Given a set of data points $x_1, \dots, x_n \in \mathbb{R}^d$ that are independently

sampled from $\lambda$, we apply Weiszfeld's algorithm for a fixed number of iterations to obtain an approximation of the geometric median $x^*$. With slight abuse of notation, we continue to denote this approximation by $x^*$. To define a distribution over the intersection point $x \in \mathbb{R}^d$, we sample from a Gaussian distribution centered at $x^*$ to ensure that sampled points remain close to the geometric median, i.e.,

$$x \sim \mathcal{N}(x^*, cI_d), \tag{14}$$

where $c > 0$ is a small constant and $I_d$ is the identity matrix. The parameter $c$ controls the concentration of the distribution around $x^*$: smaller values of $c$ produce points more tightly clustered near the geometric median, ensuring that the roots of the sampled tree systems lie close to $x^*$. The full distribution over tree systems is then defined as the joint distribution between this Gaussian distribution over root points and $k$ independent uniform distributions over directions on the unit sphere $\mathbb{S}^{d-1}$:

$$\mathcal{N}(x^*, cI_d) \otimes \mathcal{U}(\mathbb{S}^{d-1})^{\otimes k}. \tag{15}$$

**Remark 3.1.** The constant $c$ will be treated as a tuning parameter in our experiments. In practice, since the data is typically normalized, we find that setting $c = 1$ yields stable behavior and performs well across most datasets.

## 4 FERMAT-WEBER TREE-SLICED WASSERSTEIN DISTANCE

In this section, we present the Fermat–Weber Tree-Sliced Wasserstein (FW-TSW) framework and analyze its theoretical foundations along with its computational complexity.

### 4.1 FERMAT-WEBER TREE-SLICED WASSERSTEIN DISTANCE

To define the proposed discrepancy, we consider two probability measures $\mu$ and $\nu$ on $\mathbb{R}^d$ with corresponding density functions $f_\mu$ and $f_\nu$. For a given tree system $\mathcal{T} \in \mathbb{T}$ and a splitting map $\alpha$ (as in Equation (9)), we apply the Radon transform $\mathcal{R}^\alpha$ (as in Equation (4)) to obtain pushforward densities $\mathcal{R}_\mathcal{T}^\alpha f_\mu$ and $\mathcal{R}_\mathcal{T}^\alpha f_\nu$. These define new probability measures $\mu_\mathcal{T}, \nu_\mathcal{T} \in \mathcal{P}(\mathcal{T})$ supported on the tree metric space $\mathcal{T}$. The OT distance between $\mu_\mathcal{T}$ and $\nu_\mathcal{T}$ can then be computed efficiently, due to the existence of closed-form solutions for OT problems on tree spaces (Le et al., 2019).

We define our discrepancy as the expected transport cost over a distribution of tree systems. Specifically, we take the expectation of the OT distance with respect to a data-dependent sampling distribution $\sigma_{\mathrm{FW},\mu,\nu}$ (as in Equation (15)). The subscript notation $\sigma_{\mathrm{FW},\mu,\nu}$ reflects the fact that the distribution is constructed from a set of points derived from $\mu$ and $\nu$, as discussed in Section 4.2. The resulting expected transport cost defines the Fermat–Weber Tree-Sliced Wasserstein distance (FW-TSW).

**Definition 4.1** (Fermat–Weber Tree-Sliced Wasserstein distance)**.** The *Fermat–Weber Tree-Sliced Wasserstein distance* (FW-TSW), between $\mu$ and $\nu$ in $\mathcal{P}(\mathbb{R}^d)$ is defined by

$$\mathrm{FW\text{-}TSW}(\mu, \nu) := \int_\mathbb{T} \mathrm{W}_1(\mu_\mathcal{T}, \nu_\mathcal{T}) d\sigma_{\mathrm{FW},\mu,\nu}(\mathcal{T}). \tag{16}$$

### 4.2 PROPERTIES OF FERMAT-WEBER TREE-SLICED WASSERSTEIN DISTANCE

We investigate several theoretical properties of the proposed FW-TSW discrepancy. Proofs of all results presented in this section are provided in Appendix C.

**Constructing the Distribution** $\sigma_{\mathrm{FW},\mu,\nu}$**.** The sampling distribution $\sigma_{\mathrm{FW},\mu,\nu}$, as defined in Equation (15), is centered at a point $x^*$, which is the geometric median of a set of data points. We now describe how these points are constructed. In practical applications, Optimal Transport aims to align a source distribution with a target distribution—typically the observed data. Therefore, it is natural to compute $x^*$ based on samples drawn from both $\mu$ and $\nu$. Specifically, we sample $m$ points $x_1, \ldots, x_m$ from the source measure $\mu$ and $m$ points $y_1, \ldots, y_m$ from the target measure $\nu$. The point $x^*$ is then computed as the geometric median of the combined set $\{x_1, \ldots, x_m, y_1, \ldots, y_m\}$. By construction, this ensures that the distribution satisfies the symmetry property $\sigma_{\mathrm{FW},\mu,\nu} = \sigma_{\mathrm{FW},\nu,\mu}$.

**Directional Sampling.** The formulation of $\sigma_{\mathrm{FW},\mu,\nu}$ in Equation (15) also includes a directional component. Inspired by the data-dependent design of the intersection point distribution, we propose

an analogous enhancement for the directional distribution to go beyond simple uniform sampling over $\mathbb{S}^{d-1}$. To sample informative directions, we randomly select a source point $x_i$ and a target point $y_j$, and construct a direction vector as follows:

$$\theta = \left(\psi + \zeta \cdot s \cdot (x_i - y_j)\right) \Big/ \left\|\psi + \zeta \cdot s \cdot (x_i - y_j)\right\|_2 \in \mathbb{S}^{d-1}, \tag{17}$$

where $\psi \sim \mathcal{U}(\mathbb{S}^{d-1})$ is a direction sampled uniformly from $\mathbb{S}^{d-1}$; $s \sim \mathcal{U}(\{\pm 1\})$ is a random sign; $i, j \sim \mathcal{U}(\{1, \ldots, m\})$ are indices selected uniformly at random, independently; and $\zeta > 0$ is a scaling parameter that controls how strongly the direction is biased toward the vector $(x_i - y_j)$. The resulting directional distribution on $\mathbb{S}^{d-1}$ is denoted by $\sigma_{\text{dir},\mu,\nu}$. Using this, we define the enhanced sampling distribution on tree systems as:

$$\sigma^*_{\text{FW},\mu,\nu} = \mathcal{N}(x^*, I_d) \otimes (\sigma_{\text{dir},\mu,\nu})^{\otimes k}, \tag{18}$$

The resulting TSW discrepancy that uses this improved sampling strategy is defined by:

$$\text{FW-TSW}^*(\mu, \nu) \coloneqq \int_{\mathbb{T}} \text{W}_1(\mu_{\mathcal{T}}, \nu_{\mathcal{T}}) \, d\sigma^*_{\text{FW},\mu,\nu}(\mathcal{T}). \tag{19}$$

**Remark 4.2.** The random sign $s \sim \mathcal{U}(\{\pm 1\})$ in Equation (18) ensures symmetry of the directional distribution, i.e., $\sigma_{\text{dir},\mu,\nu} = \sigma_{\text{dir},\nu,\mu}$. Therefore, the sampling distribution satisfies $\sigma^*_{\text{FW},\mu,\nu} = \sigma^*_{\text{FW},\nu,\mu}$.

**Metricity of** FW-TSW. We examine whether FW-TSW satisfies the standard properties of a metric.

**Theorem 4.3.** *Both* FW-TSW *and* FW-TSW$^*$ *are semi-metrics on the space* $\mathcal{P}(\mathbb{R}^d)$. *In particular, they satisfy non-negativity, symmetry, and the identity of indiscernibles. Moreover, they satisfy the following quasi-triangle inequality: for any* $\mu_1, \mu_2, \mu_3 \in \mathcal{P}(\mathbb{R}^d)$,

$$\text{FW-TSW}(\mu_1, \mu_2) \leq \text{FW-TSW}_{\mu_1,\mu_2}(\mu_1, \mu_3) + \text{FW-TSW}_{\mu_1,\mu_2}(\mu_2, \mu_3), \tag{20}$$

*where the intermediate discrepancy term is defined as*

$$\text{FW-TSW}_{\mu_1,\mu_2}(\mu, \nu) \coloneqq \int_{\mathbb{T}} \text{W}_1(\mu_{\mathcal{T}}, \nu_{\mathcal{T}}) \, d\sigma_{\text{FW},\mu_1,\mu_2}(\mathcal{T}). \tag{21}$$

*The same properties hold for* FW-TSW$^*$.

**Invariance under Euclidean Transformations.** Since the proposed discrepancy operates on probability measures defined over $\mathbb{R}^d$, it is essential to analyze its behavior under transformations from the Euclidean group $\text{E}(d)$. For context, both the classical 2-Wasserstein distance and the Sliced $p$-Wasserstein distance are known to be invariant under Euclidean transformations. We confirm that this invariance property is preserved in our setting as well.

**Theorem 4.4.** FW-TSW *and* FW-TSW$^*$ *are invariant under Euclidean transformations on* $\mathbb{R}^d$.

**Boundedness.** We derive an upper bound related to the proposed FW-TSW discrepancy. Unlike prior TSW variants (Tran et al., 2024e; 2025a), where positional information is uncontrolled and bounds are difficult to obtain, the geometric median in FW-TSW allows for a tractable bound under mild conditions.

**Theorem 4.5.** *Let* $\mu, \nu \in \mathcal{P}(\mathbb{R}^d)$ *be two probability measures. Consider the function*

$$f(v) \coloneqq \int_{\mathbb{R}^d} \|x - v\|_2 \, d\mu(x) + \int_{\mathbb{R}^d} \|x - v\|_2 \, d\nu(x), \quad \text{for all } v \in \mathbb{R}^d, \tag{22}$$

*which associated with the joint Fermat–Weber problem of* $\mu$ *and* $\nu$. *Let* $v^* \coloneqq \arg\min_{v \in \mathbb{R}^d} f(v)$ *be the geometric median of the combined support of* $\mu$ *and* $\nu$, *and define the sampling distribution* $\bar{\sigma}_{\text{FW},\mu,\nu} \coloneqq \delta_{v^*} \otimes \mathcal{U}(\mathbb{S}^{d-1})^{\otimes k}$, *where* $\delta_{v^*}$ *is the Dirac measure centered at* $v^*$. *Then, we have:*

$$\int_{\mathbb{T}} \text{W}_1(\mu_{\mathcal{T}}, \nu_{\mathcal{T}}) \, d\bar{\sigma}_{\text{FW},\mu,\nu}(\mathcal{T}) \leq k \, \text{W}_2(\mu, \nu) + k(k-1) \cdot \frac{2\pi^{d/2}}{\Gamma\left(\frac{d+1}{2}\right)} \Gamma\left(\frac{1}{2}\right) f(v^*). \tag{23}$$

Table 1: Average Wasserstein distance between source and target distributions over 5 independent runs on the 25 Gaussians dataset. We use 100 projecting for all methods.

| Methods | Step | | | | |
|---|---|---|---|---|---|
| | 500 | 1000 | 1500 | 2000 | 2500 |
| SW Bonneel et al. (2015) | 3.65e-03 ± 1.3e-03 | 2.42e-03 ± 8.0e-04 | 2.13e-03 ± 9.0e-04 | 1.69e-03 ± 9.8e-04 | 1.01e-03 ± 9.5e-04 |
| SWGG Mahey et al. (2023) | **7.67e-04 ± 1.4e-03** | 4.85e-06 ± 5.5e-06 | 2.91e-06 ± 2.4e-06 | 2.72e-06 ± 5.3e-06 | 2.91e-06 ± 5.7e-06 |
| LCVSW Luong et al. (2024) | 1.54e-03 ± 1.1e-03 | 1.40e-03 ± 8.7e-04 | 7.84e-04 ± 5.6e-04 | 5.73e-04 ± 6.3e-04 | 6.84e-04 ± 7.9e-04 |
| TSW-SL Tran et al. (2024e) | 1.12e-03 ± 9.7e-04 | **1.37e-06 ± 8.7e-08** | 1.07e-06 ± 4.8e-08 | 9.13e-07 ± 5.2e-08 | 8.76e-07 ± 1.1e-07 |
| Db-TSW Tran et al. (2025a) | 3.42e-03 ± 7.9e-04 | 1.55e-06 ± 1.2e-07 | 1.10e-06 ± 9.2e-08 | 9.50e-07 ± 6.1e-08 | 8.55e-07 ± 5.6e-08 |
| Db-TSW$^\perp$ Tran et al. (2025a) | 2.70e-03 ± 9.0e-04 | 1.79e-06 ± 2.0e-07 | 1.25e-06 ± 9.7e-08 | 1.14e-06 ± 5.6e-08 | 1.03e-06 ± 4.8e-08 |
| FW-TSW (ours) | 2.40e-03 ± 8.9e-04 | 1.51e-06 ± 1.4e-07 | **1.03e-06 ± 1.0e-07** | 9.18e-07 ± 4.1e-08 | 8.40e-07 ± 2.6e-08 |
| FW-TSW* (ours) | 2.59e-03 ± 9.3e-04 | 1.50e-06 ± 8.9e-08 | 1.11e-06 ± 6.6e-08 | **9.04e-07 ± 1.1e-07** | **8.29e-07 ± 4.7e-08** |

**Computational Complexity.** Let $n$ and $m$ denote the number of support points in two discrete measures $\mu, \nu \in \mathcal{P}(\mathbb{R}^d)$, with $n \gg m$. The standard Sliced Wasserstein (SW) distance has a computational complexity of $\mathcal{O}(Ln \log n + Ldn)$, where $L$ is the number of random projections (Bonneel et al., 2015). More recent approaches, such as Tree-Sliced Wasserstein (TSW) and its variants TSW-SL (Tran et al., 2024e) and Db-TSW (Tran et al., 2025a), exhibit a complexity of $\mathcal{O}(Lkn \log n + Lkdn)$; here, $L$ represents the number of sampled trees and $k$ denotes the lines per tree. For FW-TSW, the complexity increases to $\mathcal{O}(Lkn \log n + Lkdn + Tnd)$, incorporating an additional $Tnd$ cost for approximating the geometric median via Weiszfeld's Algorithm (where $T$ is the maximum iterations). FW-TSW* further incurs an extra $\mathcal{O}(Lkd)$ term for generating random paths. Notably, these additional costs for FW-TSW and FW-TSW* contribute negligibly to the overall computation time, as detailed in Appendix D.1.

## 5 EXPERIMENTAL RESULTS

In this section, we present a series of experiments involving Gradient Flows, Topic Modeling and Diffusion Models to assess the effectiveness of FW-TSW and FW-TSW*. Additional experiments on point cloud and MNIST-like images are provided in D.2.

### 5.1 GRADIENT FLOW

This task employs gradient-based optimization to minimize the discrepancy between a time-evolving source distribution $\mu_t$, originating from $\mu_0$, and a fixed target distribution $\nu$. The evolution is governed by the differential equation $\partial_t \mu_t = -\nabla_{\mu_t} \mathcal{D}(\mu_t, \nu)$. In this equation, $\mathcal{D}(\mu_t, \nu)$ is a distance metric (e.g., SW, Db-TSW, or our FW-TSW and FW-TSW*).

We evaluate our proposed methods, FW-TSW and FW-TSW*, on the 25 Gaussians dataset. Table 1 presents the average Wasserstein distance between source and target distributions over five runs, using optimal learning rates for each method (details in Appendix D.2). Performance is tracked at steps 500, 1000, 1500, 2000, and 2500. While SWGG initially exhibits the lowest distance (at step 500), Db-TSW, Db-TSW$^\perp$, FW-TSW, and FW-TSW* demonstrate steady improvement, eventually outperforming SWGG. Notably, from step 2000 onwards, FW-TSW and FW-TSW* yield the best results, with FW-TSW* being the best overall at step 2500.

### 5.2 TOPIC MODELING

In this experiment, we evaluate the efficiency of our proposed TSW distance for topic modeling (Blei et al., 2003). Topic models are commonly framed as VAEs (Srivastava & Sutton, 2017), with an objective combining reconstruction and KL-divergence terms. Following Nan et al. (2019); Adhya & Sanyal (2025), we replace the KL term with our TSW objective $\inf_{\varphi, \psi} \; \mathbb{E}_{p(\mathbf{x})} \mathbb{E}_{q_\varphi(\theta|\mathbf{x})}[\mathrm{CE}(\mathbf{x}, \hat{\mathbf{x}})] + \lambda \, \mathrm{FW\text{-}TSW}(q_\varphi(\theta), p(\theta))$, where CE is the cross-entropy between original $\mathbf{x}$ and reconstruction $\hat{\mathbf{x}} = \psi(\theta)$, with encoder $\varphi$ and decoder $\psi$. We compare FW-TSW-TM and FW*-TSW-TM against SW- and TSW-based baselines. Performance is measured by topic coherence $C_\mathrm{V}$ (Röder et al., 2015). As shown in Table 2, our methods achieve higher coherence over SW and TSW variants.

### 5.3 DIFFUSION MODELS

This experiment investigates training denoising diffusion models for unconditional image synthesis. Inspired by Nguyen et al. (2024b), we integrate Wasserstein distances into the Augmented Generalized Mini-batch Energy (AGME) loss function of the Denoising Diffusion Generative Adversarial

Table 2: Average topic coherence CV across 3 datasets DBLP, M10, and BBC. Higher is better.

| Method | DBLP | M10 | BBC |
|---|---|---|---|
| LDA (Blei et al., 2003) | $0.337_{\pm 0.017}$ | $0.341_{\pm 0.018}$ | $0.457_{\pm 0.054}$ |
| ProdLDA (Srivastava & Sutton, 2017) | $0.488_{\pm 0.012}$ | $0.499_{\pm 0.020}$ | $0.688_{\pm 0.018}$ |
| WTM (Nan et al., 2019) | $0.498_{\pm 0.040}$ | $0.403_{\pm 0.047}$ | $0.741_{\pm 0.034}$ |
| SW-TM (Bonneel et al., 2015) | $0.432_{\pm 0.061}$ | $0.484_{\pm 0.043}$ | $0.760_{\pm 0.048}$ |
| RPSW-TM (Nguyen et al., 2024b) | $0.426_{\pm 0.047}$ | $0.472_{\pm 0.032}$ | $0.775_{\pm 0.026}$ |
| EBRPSW-TM (Nguyen et al., 2024b) | $0.416_{\pm 0.057}$ | $0.492_{\pm 0.054}$ | $0.777_{\pm 0.019}$ |
| TSW-SL-TM (Tran et al., 2024e) | $0.453_{\pm 0.045}$ | $0.456_{\pm 0.040}$ | $0.796_{\pm 0.038}$ |
| Db-TSW-TM (Tran et al., 2025a) | $0.441_{\pm 0.056}$ | $0.458_{\pm 0.081}$ | $0.787_{\pm 0.041}$ |
| FW-TSW-TM (ours) | $0.505_{\pm 0.056}$ | $0.498_{\pm 0.069}$ | $0.792_{\pm 0.041}$ |
| FW*-TSW-TM (ours) | $\mathbf{0.511}_{\pm 0.053}$ | $\mathbf{0.502}_{\pm 0.034}$ | $\mathbf{0.801}_{\pm 0.034}$ |

Table 3: FID scores and per-epoch training times of DDGAN variants for unconditional generation on CIFAR-10.

| Model | FID ↓ | Time/Epoch(s) ↓ |
|---|---|---|
| DDGAN Xiao et al. (2021) | 3.64 | 72 |
| SW-DD Nguyen et al. (2024b) | 2.90 | 74 |
| DSW-DD Nguyen et al. (2024b) | 2.88 | 498 |
| EBSW-DD Nguyen et al. (2024b) | 2.87 | 76 |
| RPSW-DD Nguyen et al. (2024b) | 2.82 | 76 |
| IWRPSW-DD Nguyen et al. (2024b) | 2.70 | 77 |
| TSW-SL-DD Tran et al. (2024e) | 2.83 | 80 |
| Db-TSW-DD Tran et al. (2025a) | 2.60 | 84 |
| Db-TSW-DD$^{\perp}$ Tran et al. (2025a) | 2.53 | 85 |
| FW-TSW-DD (ours) | $\underline{2.336 \pm 0.003}$ | 85 |
| FW-TSW*-DD (ours) | $\mathbf{2.315 \pm 0.002}$ | 87 |

Network (DDGAN) (Xiao et al., 2021). Our proposed methods, FW-TSW-DD and FW-TSW*-DD, are benchmarked against Sliced and Tree-Sliced Wasserstein-based DDGAN variants, with results detailed in Table 3. Details can be found in Appendix D.4.

As shown in Table 3, our proposed methods, FW-TSW-DD and FW-TSW*-DD, achieve significant FID score improvements over all baselines. Notably, they surpass the current state-of-the-art OT-based competitor, Db-TSW-DD$^{\perp}$ (Tran et al., 2025a), by substantial FID margins of $0.194$ and $0.215$, respectively. Furthermore, our methods achieve these improvements with training times comparable to existing tree-sliced techniques, highlighting their practicality for large-scale applications.

## 6 CONCLUSION

In this paper, we introduce the Fermat–Weber Tree-Sliced Wasserstein (FW-TSW) distance, a novel variant of the Tree-Sliced Wasserstein (TSW) framework inspired by the classical Fermat–Weber problem. By leveraging Weiszfeld's algorithm to sample intersection points in the tree structure, FW-TSW captures both positional and directional information through a data-dependent sampling scheme. We analyze key properties of FW-TSW, including semi-metricity, Euclidean invariance, boundedness, and computational efficiency. Empirical results on gradient flow and Diffusion Model training demonstrate improved performance with minimal overhead. A key limitation, shared with other TSW variants, is the lack of explicit transport maps. Future work may address this by developing tree-sliced frameworks that produce transport plans.

### ACKNOWLEDGMENTS

We thank the area chairs and anonymous reviewers for their comments. TL gratefully acknowledges the support of the JST-BOOST program (FY2025), JSPS KAKENHI Grant number 23K11243, and Mitsui Knowledge Industry Co., Ltd. grant. TT acknowledges support from the Application Driven Mathematics Program funded and organized by the Vingroup Innovation Fund and VinBigData.

This research / project is supported by the National Research Foundation Singapore under the AI Singapore Programme (AISG Award No: AISG2-TC-2023-012-SGIL). This research / project is supported by the Ministry of Education, Singapore, under the Academic Research Fund Tier 1 (FY2023) (A-8002040-00-00, A-8002039-00-00). This research / project is also supported by the NUS Presidential Young Professorship Award (A-0009807-01-00) and the NUS Artificial Intelligence Institute–Seed Funding (A-8003062-00-00).

**Ethics Statement.** Given the nature of the work, we do not foresee any negative societal and ethical impacts of our work.

**Reproducibility Statement.** Source codes for our experiments are provided in the supplementary materials of the paper. The details of our experimental settings and computational infrastructure are given in Section D and the Appendix. All datasets that we used in the paper are published, and they are easy to access in the Internet.

**LLM Usage Declaration.** We use large language models (LLMs) for grammar checking and correction.

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

## TABLE OF NOTATION

| | |
|---|---|
| $\mathbb{R}^d$ | $d$-dimensional Euclidean space |
| $\|\cdot\|_2$ | Euclidean norm |
| $\mathbb{S}^{d-1}$ | $(d-1)$-dimensional hypersphere |
| $\theta, \psi$ | unit vector |
| $\sqcup$ | disjoint union |
| $L^1(X)$ | space of Lebesgue integrable functions on $X$ |
| $\mathcal{P}(X)$ | space of probability measures on $X$ |
| $\mathcal{M}(X)$ | space of measures on $X$ |
| $\mu, \nu$ | measures |
| $\delta(\cdot)$ | 1-dimensional Dirac delta function |
| $\mathcal{U}(\mathbb{S}^{d-1})$ | uniform distribution on $\mathbb{S}^{d-1}$ |
| $\mathcal{C}(X,Y)$ | space of continuous maps from $X$ to $Y$ |
| $d(\cdot,\cdot)$ | metric in metric space |
| $d_{\mathcal{T}}(\cdot,\cdot)$ | tree metric |
| $\mathrm{E}(d)$ | Euclidean group of order $d$ |
| $\mathrm{W}_p$ | $p$-Wasserstein distance |
| $\mathrm{SW}_p$ | Sliced $p$-Wasserstein distance |
| $\Lambda$ | (rooted) subtree |
| $\mathcal{T}$ | tree system |
| $L$ | number of Monte Carlo samples |
| $k$ | number of lines in a system of lines or a tree system |
| $\mathcal{R}^\alpha$ | Radon Transform on Systems of Lines |
| $\Delta_{k-1}$ | $(k-1)$-dimensional standard simplex |
| $\alpha$ | splitting map |
| $\xi, \zeta, c$ | tuning parameter |
| $\mathbb{T}$ | space of tree systems |
| $\sigma, \sigma_{\mathrm{FW}}, \bar{\sigma}_{\mathrm{FW}}, \sigma_{\mathrm{FW}}^*, \sigma_{\mathrm{dir}}$ | distributions on (components of) space of tree systems |
| $\mathcal{N}$ | normal (Gaussian) distribution |
| $\mathcal{U}$ | uniform distribution |
| $\delta$ | Dirac delta distribution |
| $\varepsilon$ | threshold in Weiszfeld's algorithm. |

# Appendix of "Revisiting Tree-Sliced Wasserstein Distance Through the Lens of the Fermat–Weber Problem"

**Table of Contents**

## A  BACKGROUND ON TREE-SLICED WASSERSTEIN DISTANCE IN EUCLIDEAN SPACES

This section revisits the fundamental components of the Tree-Sliced Wasserstein (TSW) distance, formulated over tree systems embedded in Euclidean spaces. For completeness, we summarize key definitions and core mathematical formulations. Readers are referred to Tran et al. (2024e; 2025a) for detailed proofs and extended exposition.

### A.1  TREE SYSTEM CONSTRUCTION

A line in $\mathbb{R}^d$ is represented as a tuple $(x, \theta) \in \mathbb{R}^d \times \mathbb{S}^{d-1}$, where $x$ denotes a reference point and $\theta$ is a direction. The line is parameterized by $x + t \cdot \theta$ for $t \in \mathbb{R}$. We denote a line by $l = (x_l, \theta_l)$. A point on this line is written either as $(x, l)$ or as $(t_x, l)$, depending on whether we refer to the point in ambient space or to its parametrization along $l$, respectively. A system of $k$ lines in $\mathbb{R}^d$ is an element of the product space $(\mathbb{R}^d \times \mathbb{S}^{d-1})^k$, abbreviated as $\mathbb{T}$. An element $\mathcal{T} \in \mathbb{T}$ denotes a specific configuration of $k$ lines. A line system $\mathcal{T}$ is *connected* if the union of all lines in $\mathcal{T}$ forms a connected set in $\mathbb{R}^d$. A tree structure can be enforced by removing selected intersection points, so that any two points on the resulting configuration are connected by a unique path. The term *tree system* reflects the property that any two points are connected via a unique path, akin to trees in graph theory. Using preserved intersections, we build a topological tree system by coherently gluing segments of $\mathbb{R}$ via disjoint union and quotient topology (Hatcher, 2005), resulting in a space endowed with a valid tree metric.

## A.2  A GENERALIZED RADON TRANSFORM OVER LINE SYSTEMS

Let $L^1(\mathbb{R}^d)$ denote the space of integrable functions on $\mathbb{R}^d$. Given a system of lines $\mathcal{T} \in \mathbb{T}$, define $L^1(\mathcal{T})$ as the space of functions $f$ such that $\|f\|_{\mathcal{T}} = \sum_{l \in \mathcal{T}} \int_{\mathbb{R}} |f(t_x, l)| \, dt_x < \infty$.

The $(k-1)$-dimensional simplex is defined as $\Delta_{k-1} = \{(a_l)_{l \in \mathcal{T}} \in \mathbb{R}^k \mid a_l \geq 0, \ \sum_{l \in \mathcal{T}} a_l = 1\}$. Let $\mathcal{C}(\mathbb{R}^d \times \mathbb{T}, \Delta_{k-1})$ denote the space of continuous functions, called *splitting maps*, from $\mathbb{R}^d \times \mathbb{T}$ to $\Delta_{k-1}$. Given a splitting map $\alpha$ and $f \in L^1(\mathbb{R}^d)$, define the projection operator:

$$\mathcal{R}_{\mathcal{T}}^{\alpha} f(x, l) = \int_{\mathbb{R}^d} f(y) \cdot \alpha(y, \mathcal{T})_l \cdot \delta\left(t_x - \langle y - x_l, \theta_l \rangle\right) dy, \tag{24}$$

where $(x_l, \theta_l)$ specifies line $l$ and $\delta$ is the Dirac delta. This operator maps $f$ to a function on $\mathcal{T}$, the union of the lines. Extending over all $\mathcal{T} \in \mathbb{T}$, define the *Radon Transform on Tree Systems* by

$$\mathcal{R}^{\alpha} : L^1(\mathbb{R}^d) \longrightarrow \prod_{\mathcal{T} \in \mathbb{T}} L^1(\mathcal{T}), \quad f \longmapsto (\mathcal{R}_{\mathcal{T}}^{\alpha} f)_{\mathcal{T} \in \mathbb{T}}. \tag{25}$$

If $\alpha$ is invariant under the Euclidean group $\mathrm{E}(d)$, then $\mathcal{R}^{\alpha}$ is injective.

## A.3  TREE-SLICED WASSERSTEIN DISTANCE IN EUCLIDEAN SPACES

Let $\mu, \nu \in \mathcal{P}(\mathbb{R}^d)$ be probability measures. For a tree-structured line system $\mathcal{T} \in \mathbb{T}$ and an $\mathrm{E}(d)$-invariant splitting map $\alpha$, let $\mathcal{R}_{\mathcal{T}}^{\alpha} \mu$ and $\mathcal{R}_{\mathcal{T}}^{\alpha} \nu$ be the pushforwards of $\mu$ and $\nu$, respectively. Equipped with the tree metric $d_{\mathcal{T}}$, we compute the 1-Wasserstein distance:

$$\mathrm{W}_{d_{\mathcal{T}}, 1}(\mathcal{R}_{\mathcal{T}}^{\alpha} \mu, \mathcal{R}_{\mathcal{T}}^{\alpha} \nu). \tag{26}$$

The *Tree-Sliced Wasserstein* (TSW) distance (Tran et al., 2025a) is defined as:

$$\mathrm{TSW}(\mu, \nu) \coloneqq \int_{\mathbb{T}} \mathrm{W}_{d_{\mathcal{T}}, 1}(\mathcal{R}_{\mathcal{T}}^{\alpha} \mu, \mathcal{R}_{\mathcal{T}}^{\alpha} \nu) \, d\sigma(\mathcal{T}), \tag{27}$$

where $\sigma$ is a probability distribution over $\mathbb{T}$. Though the notation omits explicit dependence on $\alpha$, $\mathbb{T}$, and $\sigma$, the metric depends on all three.

**Remark A.1.** If tree systems are reduced to single lines, TSW recovers the classical Sliced Wasserstein distance.

$\mathrm{E}(d)$**-Invariant Splitting Maps.**  Let $x \in \mathbb{R}^d$ and $\mathcal{T} \in \mathbb{T}$. Define the Euclidean distance from $x$ to line $l$ as $d(x, \mathcal{T})_l = \inf_{y \in l} \|x - y\|_2$. This function is invariant under $\mathrm{E}(d)$. A practical choice for $\alpha$ is the softmax:

$$\alpha(x, \mathcal{T}) = \mathrm{softmax}(\{\xi \cdot d(x, \mathcal{T})_l\}_{l \in \mathcal{T}}), \tag{28}$$

with $\xi > 0$ controlling the sharpness of the distribution over lines.

**Remark A.2.** Equivariant neural networks (Cohen & Welling, 2016) hard-code task symmetries into model architectures, yielding stronger generalization and improved sample efficiency. They have demonstrated broad empirical effectiveness in trajectory prediction (Walters et al., 2020), robotics (Simeonov et al., 2022), graph learning (Satorras et al., 2021; Tran et al., 2024b), Optimal Transport–based approaches (Pham et al., 2026; Tran et al., 2026b; 2025e; 2024f; 2025c), equivariant metanetworks (Tran et al., 2026a; Vo et al., 2025; Tran et al., 2024a; 2025b), and weight-space analysis (Tran et al., 2025f). Overall, exploiting equivariance systematically improves accuracy, data efficiency, and robustness to distributional shifts.

## B  GENERAL FORMULATION FOR THE FERMAT-WEBER PROBLEM

In this section, we provide the background on the Fermat–Weber Problem and Weiszfeld's Algorithm. We begin with the continuous formulation of the Fermat–Weber problem:

$$v^* = \arg \min_{v \in \mathbb{R}^d} \int_{\mathbb{R}^d} \|x - v\|_2 \, d\lambda(x), \tag{29}$$

**Discrete Approximation via Monte Carlo.** Let $\{x^j\}_{j=1}^n$ be i.i.d. samples from $\lambda$, with associated weights $\lambda(x^j)$. A Monte Carlo approximation of Equation (29) is

$$v^* \approx \arg\min_{v \in \mathbb{R}^d} \sum_{j=1}^n \lambda(x^j) \|x^j - v\|_2. \tag{30}$$

Define the *weighted geometric median objective*

$$F(v) = \sum_{j=1}^n \lambda(x^j) \|x^j - v\|_2.$$

Since $F(v)$ is convex (but non-differentiable at the sample points), we employ the classic Weiszfeld iteration to compute its minimizer.

**Weiszfeld's Iteration.** Denote by $v^{(t)} \in \mathbb{R}^d$ the estimate at iteration $t$. The standard update is

$$v^{(t+1)} = T(v^{(t)}) = \left( \sum_{j=1}^n \frac{\lambda(x^j)\, x^j}{\|v^{(t)} - x^j\|_2} \right) \Big/ \left( \sum_{j=1}^n \frac{\lambda(x^j)}{\|v^{(t)} - x^j\|_2} \right), \tag{31}$$

provided $v^{(t)} \neq x^j$ for all $j$. If $v^{(t)} = x^k$ for some index $k$, one may either terminate (since $F$ is minimized at that sample) or apply the more robust variant below.

**Handling Coincident Iterates.** To avoid the "sticking" phenomenon when $v^{(t)}$ exactly equals a sample $x^k$, one introduces a small perturbation or uses the following safeguarded map:

$$\tilde{T}(v) = \begin{cases} \dfrac{\left( \sum_{j=1}^n \frac{\lambda(x^j)\, x^j}{\|v - x^j\|_2} \right)}{\left( \sum_{j=1}^n \frac{\lambda(x^j)}{\|v - x^j\|_2} \right)}, & v \neq x^1, \dots, x^n, \\[2ex] x^k, & v = x^k, \quad \text{provided } \sum_{j \neq k} \frac{\lambda(x^j)}{\|x^k - x^j\|_2} = 0, \end{cases} \tag{32}$$

and then define

$$v^{(t+1)} = (1 - \beta(v^{(t)}))\, \tilde{T}(v^{(t)}) + \beta(v^{(t)})\, v^{(t)}, \tag{33}$$

where $\beta(v) \in [0, 1]$ is chosen to ensure descent in $F$.

**Convergence and Remarks.**

- Under mild conditions (no three points colinear, positive weights), the sequence $\{v^{(t)}\}$ converges to the unique geometric median (Weiszfeld & Plastria, 2009).
- In practice, when the Monte Carlo weights $\lambda(x^j)$ are noisy, Weiszfeld's algorithm can be sensitive. In such cases, gradient-based methods (e.g. subgradient descent) with a suitable smoothing may be preferred.
- Weiszfeld's iteration typically converges in $\mathcal{O}(1/t)$ rate and is computationally inexpensive per step, making it effective when the sample size $n$ is moderate.

## C  THEORETICAL PROOFS

In this section, we provide proofs for all results stated in the paper.

### C.1  PROOF FOR THEOREM 4.3

*Proof.* We recall the Fermat–Weber Tree-Sliced Wasserstein distance between $\mu$ and $\nu$ in $\mathcal{P}(\mathbb{R}^d)$ is defined by

$$\text{FW-TSW}(\mu, \nu) = \int_{\mathbb{T}} W_1(\mu_{\mathcal{T}}, \nu_{\mathcal{T}}) d\sigma_{\text{FW}, \mu, \nu}(\mathcal{T}). \tag{34}$$

**Non-negativity.** Since $W_1$ is a valid distance, we have $W_1(\mu_\mathcal{T}, \nu_\mathcal{T}) \geq 0$ for all $\mathcal{T} \in \mathbb{T}$. It implies that

$$\text{FW-TSW}(\mu, \nu) = \int_\mathbb{T} W_1(\mu_\mathcal{T}, \nu_\mathcal{T}) d\sigma_{\text{FW},\mu,\nu}(\mathcal{T}) \geq 0. \tag{35}$$

**Symmetry.** Given $\mu, \nu \in \mathcal{P}(\mathbb{R})$. By design, we $d\sigma_{\text{FW},\mu,\nu} = d\sigma_{\text{FW},\nu,\mu}$. Moreover, since $W_1$ is a valid distance, we have $W_1(\mu_\mathcal{T}, \nu_\mathcal{T}) = W_1(\nu_\mathcal{T}, \mu_\mathcal{T})$. It implies that

$$\begin{aligned}
\text{FW-TSW}(\mu, \nu) &= \int_\mathbb{T} W_1(\mu_\mathcal{T}, \nu_\mathcal{T}) d\sigma_{\text{FW},\mu,\nu}(\mathcal{T}) \\
&= \int_\mathbb{T} W_1(\nu_\mathcal{T}, \mu_\mathcal{T}) d\sigma_{\text{FW},\nu,\mu}(\mathcal{T}) = \text{FW-TSW}(\nu, \mu). \tag{36}
\end{aligned}$$

**The identity of indiscernibles.** Assume that $\text{FW-TSW}(\mu, \nu) = 0$. Since $\sigma_{\text{FW},\nu,\mu}$ is a continuous distribution on $\mathbb{T}$, we have $W_1(\nu_\mathcal{T}, \mu_\mathcal{T}) = 0$ for all $\mathcal{T} \in \mathbb{T}$. Since the Radon Transform on Tree Systems is injective (refer to Tran et al. (2025a)), it implies that $\nu = \mu$.

**Quasi Triangular Inequality.** For $\mu_1, \mu_2, \mu_3 \in \mathcal{P}(\mathbb{R}^d)$, we have

$$\begin{aligned}
\text{FW-TSW}(\mu_1, \mu_2) &= \text{FW-TSW}_{\mu_1,\mu_2}(\mu_1, \mu_2) \\
&= \int_\mathbb{T} W_1((\mu_1)_\mathcal{T}, (\mu_2)_\mathcal{T}) d\sigma_{\text{FW},\mu_1,\mu_2}(\mathcal{T}) \\
&\leq \int_\mathbb{T} W_1((\mu_1)_\mathcal{T}, (\mu_3)_\mathcal{T}) + W_1((\mu_2)_\mathcal{T}, (\mu_3)_\mathcal{T}) d\sigma_{\text{FW},\mu_1,\mu_2}(\mathcal{T})+ \\
&= \int_\mathbb{T} W_1((\mu_1)_\mathcal{T}, (\mu_3)_\mathcal{T}) d\sigma_{\text{FW},\mu_1,\mu_2}(\mathcal{T}) + \int_\mathbb{T} W_1((\mu_2)_\mathcal{T}, (\mu_3)_\mathcal{T}) d\sigma_{\text{FW},\mu_1,\mu_2}(\mathcal{T}) \\
&= \text{FW-TSW}_{\mu_1,\mu_3}(\mu_1, \mu_2) + \text{FW-TSW}_{\mu_2,\mu_3}(\mu_1, \mu_2). \tag{37}
\end{aligned}$$

Therefore, the proof is completed. $\square$

## C.2 PROOF FOR THEOREM 4.4

*Proof.* Note that, the Fermat-Weber problem preserves Euclidean transformations, which means

$$\begin{aligned}
\operatorname*{argmin}_{x \in \mathbb{R}^d} \int_{\mathbb{R}^d} \|x - y\|_2 \, d(g\sharp\lambda)(y) &= \operatorname*{argmin}_{x \in \mathbb{R}^d} \int_{\mathbb{R}^d} \|g^{-1}x - g^{-1}y\|_2 \, d(g\sharp\lambda)(y) \\
&= \operatorname*{argmin}_{x \in \mathbb{R}^d} \int_{\mathbb{R}^d} \|g^{-1}x - y\|_2 \, d\lambda(y) \\
&= g\left(\operatorname*{argmin}_{x \in \mathbb{R}^d} \int_{\mathbb{R}^d} \|x - y\|_2 \, d\lambda(y)\right). \tag{38}
\end{aligned}$$

It implies that $\sigma_{\text{FW},\mu,\nu}$ is equivariant under the action of Euclidean group. Not that, since the Radon Transform on Tree Systems is also equivariant under the action of Euclidean group, it leads to the induce distance FW-TSW is invariant.

Recall the construction of the distribution $\sigma_{\text{dir},\mu,\nu}$ as follows:

$$\theta = \left(\psi + \zeta \cdot s \cdot (x_i - y_j)\right)\Big/\left\|\psi + \zeta \cdot s \cdot (x_i - y_j)\right\|_2 \in \mathbb{S}^{d-1}, \tag{39}$$

where $\psi \sim \mathcal{U}(\mathbb{S}^{d-1})$ is a direction sampled uniformly from $\mathbb{S}^{d-1}$; $s \sim \mathcal{U}(\{\pm 1\})$ is a random sign; $i, j \sim \mathcal{U}(\{1, \ldots, m\})$ are indices selected uniformly at random, independently; and $\zeta > 0$ is a scaling parameter that controls how strongly the direction is biased toward the vector $(x_i - y_j)$. By design, $\sigma_{\text{dir},\mu,\nu}$ is equivariant. Thus, the induce distribution $\sigma^*_{\text{FW},\mu,\nu}$ on $\mathbb{T}$ is equivariant. By the same argument as above, FW-TSW$^*$ is invariant. $\square$

### C.3  PROOF FOR THEOREM 4.5

*Proof.* Following the definition of Tree-Sliced Wasserstein, we can write

$$W_1(\mu_\mathcal{T}, \nu_\mathcal{T}) = \inf_{\pi \in \tilde{P}} \int_{\mathcal{T} \times \mathcal{T}} d_\mathcal{T}(x, y) \pi(x, y), \tag{40}$$

where $\tilde{P}$ denotes the set of couplings on $\mathcal{P}(\mathcal{T}) \times \mathcal{P}(\mathcal{T})$ and $d_\mathcal{T}$ denotes the distance between two nodes on the tree system $\mathcal{T}$. It should be further noted that, due to the way we have constructed $\mathcal{T}$, the distance between two nodes can be explicitly calculated. More specifically, the formula for $d_\mathcal{T}$ can be derived as follow

$$d_\mathcal{T}(x, y) = \begin{cases} |x^\top \theta_i - y^\top \theta_i| & \text{for x,y belongs to the same edge } \theta_i \\ |x^\top \theta_i - v^\top \theta_i| + |y^\top \theta_j - v^\top \theta_j| & \text{for x,y belongs to edge } \theta_i \text{ and } \theta_j, \text{ respectively} \end{cases} \tag{41}$$

Where $v$ is denoted as the only vertex of the tree system.

Next, we derive an inequality that upper bounds the infimum over couplings on the tree space defined in Equation (40) by an infimum over couplings on $\mathbb{R}^d$, thereby simplifying the problem and enabling more tractable analysis in subsequent step.

**Claim 1.** Denote $\tilde{R}$ as the set of couplings on $\mathbb{R}^d \times \mathbb{R}^d$, we show that

$$\inf_{\pi \in \tilde{P}} \int_{\mathcal{T} \times \mathcal{T}} d_\mathcal{T}(x, y) \pi(x, y)$$

$$\leq \inf_{\tau \in \tilde{R}} \int_{\mathbb{R}^d \times \mathbb{R}^d} \left[ \sum_{i=1}^{k} |x^\top \theta_i - y^\top \theta_i| + \sum_{\substack{i \neq j \\ i,j \in [k]}} \left( |x^\top \theta_i - \bar{v}_{i,\mathcal{L}}| + |y^\top \theta_j - \bar{v}_{j,\mathcal{L}}| \right) \right] d\tau(x, y). \tag{42}$$

*Proof for Claim 1.* From the formula of $d_\mathcal{T}(x, y)$ in Equation (41). One can show that

$$d_\mathcal{T}(x, y) \leq \sum_{i=1}^{k} |x^\top \theta_i - y^\top \theta_i| + \sum_{\substack{i \neq j \\ i,j \in [k]}} \left( |x^\top \theta_i - \bar{v}_{i,\mathcal{L}}| + |y^\top \theta_j - \bar{v}_{j,\mathcal{L}}| \right) \quad \forall x, y \in \mathbb{R}^d$$

Hence, for every $\tau \in \tilde{R}$, there exist $\pi \in \tilde{P}$ such that

$$\int_{\mathcal{T} \times \mathcal{T}} d_\mathcal{T}(x, y) \pi(x, y)$$

$$\leq \int_{\mathbb{R}^d \times \mathbb{R}^d} \left[ \sum_{i=1}^{k} |x^\top \theta_i - y^\top \theta_i| + \sum_{\substack{i \neq j \\ i,j \in [k]}} \left( |x^\top \theta_i - \bar{v}_{i,\mathcal{L}}| + |y^\top \theta_j - \bar{v}_{j,\mathcal{L}}| \right) \right] d\tau(x, y) \tag{43}$$

Thus, if we denote $\bar{\tau}$ as the solution to the right-hand side infimum of **Claim 1**, it implies

$$\inf_{\pi \in \tilde{P}} \int_{\mathcal{T} \times \mathcal{T}} d_\mathcal{T}(x, y) \pi(x, y)$$

$$\leq \int_{\mathcal{T} \times \mathcal{T}} d_\mathcal{T}(x, y) \bar{\pi}(x, y)$$

$$\leq \int_{\mathbb{R}^d \times \mathbb{R}^d} \left[ \sum_{i=1}^{k} |x^\top \theta_i - y^\top \theta_i| + \sum_{\substack{i \neq j \\ i,j \in [k]}} \left( |x^\top \theta_i - \bar{v}_{i,\mathcal{L}}| + |y^\top \theta_j - \bar{v}_{j,\mathcal{L}}| \right) \right] d\bar{\tau}(x, y)$$

$$= \inf_{\tau \in \tilde{R}} \int_{\mathbb{R}^d \times \mathbb{R}^d} \left[ \sum_{i=1}^{k} |x^\top \theta_i - y^\top \theta_i| + \sum_{\substack{i \neq j \\ i,j \in [k]}} \left( |x^\top \theta_i - \bar{v}_{i,\mathcal{L}}| + |y^\top \theta_j - \bar{v}_{j,\mathcal{L}}| \right) \right] d\tau(x, y). \tag{44}$$

This completes the proof for **Claim 1**.

Additionally, since $\tau$ is a coupling on $\mathbb{R}^d \times \mathbb{R}^d$, the right hand side of **Claim 1** can be further simplified as follow

$$
\inf_{\tau \in \tilde{R}} \int_{\mathbb{R}^d \times \mathbb{R}^d} \left[ \sum_{i=1}^{k} |x^\top \theta_i - y^\top \theta_i| + \sum_{\substack{i \neq j \\ i,j \in [k]}} \left( |x^\top \theta_i - \bar{v}_{i,\mathcal{L}}| + |y^\top \theta_j - \bar{v}_{j,\mathcal{L}}| \right) \right] d\tau(x,y)
$$

$$
= \inf_{\tau \in \tilde{R}} \left\{ \mathcal{A}_0 + \int_{\mathbb{R}^d \times \mathbb{R}^d} \left[ \sum_{\substack{i \neq j \\ i,j \in [k]}} \left( |x^\top \theta_i - v_{\mathcal{L}}^\top \theta_i| + |y^\top \theta_j - v_{\mathcal{L}}^\top \theta_j| \right) \right] d\tau(x,y) \right\}
$$

$$
= \inf_{\tau \in \tilde{R}} \left\{ \mathcal{A}_0 + (k-1) \sum_{i=1}^{k} \int_{\mathbb{R}^d \times \mathbb{R}^d} \left[ |x^\top \theta_i - v_{\mathcal{L}}^\top \theta_i| + |y^\top \theta_j - v_{\mathcal{L}}^\top \theta_j| \right] d\tau(x,y) \right\}
$$

$$
= \inf_{\tau \in \tilde{R}} \left\{ \mathcal{A}_0 + (k-1) \sum_{i=1}^{k} \left\{ \left[ \int_{\mathbb{R}^d \times \mathbb{R}^d} |x^\top \theta_i - v_{\mathcal{L}}^\top \theta_i| d\tau(x,y) \right] + \left[ \int_{\mathbb{R}^d \times \mathbb{R}^d} |y^\top \theta_j - v_{\mathcal{L}}^\top \theta_j| d\tau(x,y) \right] \right\} \right\}
$$

$$
= \inf_{\tau \in \tilde{R}} \left\{ \mathcal{A}_0 + (k-1) \sum_{i=1}^{k} \left\{ \left[ \int_{\mathbb{R}^d} |x^\top \theta_i - v_{\mathcal{L}}^\top \theta_i| d\mu(x) \right] + \left[ \int_{\mathbb{R}^d} |y^\top \theta_j - v_{\mathcal{L}}^\top \theta_j| d\nu(y) \right] \right\} \right\}
$$

$$
= \inf_{\tau \in \tilde{R}} (\mathcal{A}_0) + (k-1) \sum_{i=1}^{k} \left\{ \left[ \int_{\mathbb{R}^d} |x^\top \theta_i - v_{\mathcal{L}}^\top \theta_i| d\mu(x) \right] + \left[ \int_{\mathbb{R}^d} |y^\top \theta_j - v_{\mathcal{L}}^\top \theta_j| d\nu(y) \right] \right\},
$$

where

$$
\mathcal{A}_0 := \int_{\mathbb{R}^d \times \mathbb{R}^d} \left[ \sum_{i=1}^{k} |x^\top \theta_i - y^\top \theta_i| \right] d\tau(x,y). \tag{45}
$$

Now, by applying the above calculations, **Claim 1** and Equation (40) we can derive an upper bound for FW-TSW as below

$$
\int_{\mathbb{T}} W_1(\mu_{\mathcal{T}}, \nu_{\mathcal{T}}) \, d\bar{\sigma}_{\text{FW},\mu,\nu}(\mathcal{T})
$$

$$
\leq \int_{\mathbb{T}} \left\{ \inf_{\tau \in \tilde{R}} \int_{\mathbb{R}^d \times \mathbb{R}^d} \left[ \sum_{i=1}^{k} |x^\top \theta_i - y^\top \theta_i| \right] d\tau(x,y) \right\} d\bar{\sigma}_{\text{FW},\mu,\nu}(\mathcal{T})
$$

$$
+ (k-1) \sum_{i=1}^{k} \left\{ \left[ \int_{\mathbb{R}^d \times \mathcal{T}} |x^\top \theta_i - v^\top \theta_i| d(\mu \times \sigma_v)(x \times v) \right] \right.
$$

$$
\left. + \left[ \int_{\mathbb{R}^d \times \mathcal{T}} |y^\top \theta_i - v^\top \theta_i| d(\nu \times \sigma_v)(y \times v) \right] \right\} \tag{46}
$$

To further upper bound Equation (46), we introduce the following two claims, each providing a bound for one of the terms in Equation (46).

**Claim 2.** It can be shown that

$$
\int_{\mathbb{T}} \left\{ \inf_{\tau \in \tilde{R}} \int_{\mathbb{R}^d \times \mathbb{R}^d} \left[ \sum_{i=1}^{k} |x^\top \theta_i - y^\top \theta_i| \right] d\tau(x,y) \right\} d\bar{\sigma}_{\text{FW},\mu,\nu}(\mathcal{T}) \leq k \, W_2(\mu, \nu) \tag{47}
$$

*Proof for Claim 2.* It is trivial to see that $|x^\top \theta_i - y^\top \theta_i| \leq \|x - y\|_2$ for all $i$. Thus,

$$
\inf_{\tau \in \tilde{R}} \int_{\mathbb{R}^d \times \mathbb{R}^d} \left[ \sum_{i=1}^{k} |x^\top \theta_i - y^\top \theta_i| \right] d\tau(x,y)
$$

$$
\leq k \inf_{\tau \in \tilde{R}} \int_{\mathbb{R}^d \times \mathbb{R}^d} \|x - y\|_2 d\tau(x,y) = k W_2(\mu, \nu) \tag{48}
$$

This directly lead to

$$\int_{\mathbb{T}} \left\{ \inf_{\tau \in \tilde{R}} \int_{\mathbb{R}^d \times \mathbb{R}^d} \left[ \sum_{i=1}^{k} |x^\top \theta_i - y^\top \theta_i| \right] d\tau(x,y) \right\} d\bar{\sigma}_{\text{FW},\mu,\nu}(\mathcal{T}) \le k \, W_2(\mu, \nu) \qquad (49)$$

which completes the proof for **Claim 2**.

**Claim 3.** It can be proven that

$$(k-1) \sum_{i=1}^{k} \left\{ \left[ \int_{\mathbb{R}^d \times \mathcal{T}} |x^\top \theta_i - v^\top \theta_i| d(\mu \times \sigma_v)(x \times v) \right] \right.$$
$$\left. + \left[ \int_{\mathbb{R}^d \times \mathcal{T}} |y^\top \theta_i - v^\top \theta_i| d(\nu \times \sigma_v)(y \times v) \right] \right\} = Cf(v^*), \qquad (50)$$

where

$$C = k(k-1) \cdot \frac{2\pi^{d/2}}{\Gamma\left(\dfrac{d+1}{2}\right)} \Gamma\left(\frac{1}{2}\right)$$

*Proof for Claim 3.* We first do the following transformations to the left-hand side of **Claim 3**,

$$\mathcal{A}_2 := (k-1) \sum_{i=1}^{k} \left\{ \left[ \int_{\mathbb{R}^d \times \mathcal{T}} |x^\top \theta_i - v^\top \theta_i| d(\mu \times \sigma_v)(x \times v) \right] \right.$$
$$\left. + \left[ \int_{\mathbb{R}^d \times \mathcal{T}} |y^\top \theta_i - v^\top \theta_i| d(\nu \times \sigma_v)(y \times v) \right] \right\}$$
$$= (k-1)k \int_{\mathbb{R}^d} \left( \int_{\mathbb{S}^{d-1}} \left( \int_{\mathbb{R}^d} |x^\top \theta - v^\top \theta| d\mu(x) \right. \right.$$
$$\left. \left. + \int_{\mathbb{R}^d} |x^\top \theta - v^\top \theta| d\nu(x) \right) d\theta \right) d\sigma_v(v)$$
$$= (k-1)k \int_{\mathbb{R}^d} \left( \int_{\mathbb{R}^d} \left( \int_{\mathbb{S}^{d-1}} |x^\top \theta - v^\top \theta| d\theta \right) d\mu(x) \right.$$
$$\left. + \int_{\mathbb{R}^d} \left( \int_{\mathbb{S}^{d-1}} |x^\top \theta - v^\top \theta| d\theta \right) d\nu(x) \right) d\sigma_v(v) \qquad (51)$$

Note that, for an $u \in \mathbb{R}^d$, one has:

$$\int_{\mathbb{S}^{d-1}} |u\theta^\top| d\theta = \frac{2\pi^{d/2}}{\Gamma\left(\frac{d+1}{2}\right)} \Gamma\left(\frac{1}{2}\right) \cdot \|u\|_2 \qquad (52)$$

It implies that:

$$\mathcal{A}_2 = k(k-1) \cdot \frac{2\pi^{d/2}}{\Gamma\left(\frac{d+1}{2}\right)} \Gamma\left(\frac{1}{2}\right) \cdot \int_{\mathbb{R}^d} \left( \int_{\mathbb{R}^d} \|x - v\|_2 d\mu(x) + \int_{\mathbb{R}^d} \|y - v\|_2 d\nu(y) \right) d\sigma_v(v)$$
$$= Cf(v^*). \qquad (53)$$

This completes the proof of **Claim 3**.

By combining **Claim 1**, **Claim 2** and **Claim 3** in addition with the primal formula in Equation (40), we yield the result stated in Theorem 4.5. $\qquad \square$

## D  EXPERIMENTAL DETAILS

### D.1  RUNTIME AND MEMORY ANALYSIS

We analyze the computational and memory complexity of the most expensive operations in our proposed distance measures, as summarized in Table 4. We also compare the runtime of our method

Table 4: Complexity Analysis of FW-TSW and FW-TSW$^*$.

| Distance | Operation | Description | Computation | Memory |
|---|---|---|---|---|
| FW-TSW | Projection | Matrix multiplication of points and lines | $O(Lknd)$ | $O(Lkd + nd)$ |
| | Distance-based weight splitting | Distance calculation and softmax | $O(Lknd)$ | $O(Lkn + Lkd + nd)$ |
| | Sorting | Sorting projected coordinates | $O(Lkn \log n)$ | $O(Lkn)$ |
| | Weiszfeld's Algorithm | Approximating geometric median | $O(Tnd)$ | $O(Tnd)$ |
| | **Total** | | $O(Lknd + Lkn \log n + Tnd)$ | $O(Lkn + Lkd + nd + Tnd)$ |
| FW-TSW$^*$ | Projection | Matrix multiplication of points and lines | $O(Lknd)$ | $O(Lkd + nd)$ |
| | Distance-based weight splitting | Distance calculation and softmax | $O(Lknd)$ | $O(Lkn + Lkd + nd)$ |
| | Sorting | Sorting projected coordinates | $O(Lkn \log n)$ | $O(Lkn)$ |
| | Weiszfeld's Algorithm | Approximating geometric median | $O(Tnd)$ | $O(Tnd)$ |
| | Generating paths | Generating random paths | $O(Lkd)$ | $O(Lkd)$ |
| | **Total** | | $O(Lknd + Lkn \log n + Tnd)$ | $O(Lkn + Lkd + nd + Tnd)$ |

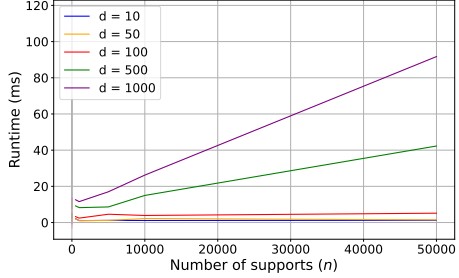 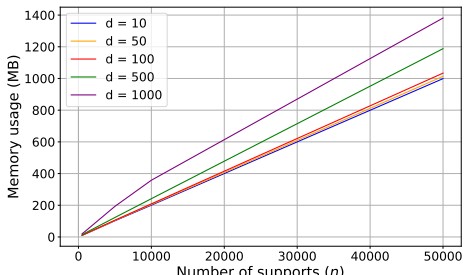

Figure 2: Execution time and memory usage of FW-TSW.

with other approaches. In Figure 4, we use $L = 2500$ trees and $k = 4$ lines for the Tree Sliced Wasserstein variants. For the Sliced Wasserstein (SW) method, we set $L = 10000$ projections.

Next, we analyze the runtime and memory performance of FW-TSW and FW-TSW$^*$ by varying $n$ and $d$, using a single NVIDIA H100 GPU. We use $L = 2500$ trees and $k = 4$ lines for all runs. We select $n \in \{500, 1000, 5000, 10000, 50000\}$ and $d \in \{10, 50, 100, 500, 1000\}$.

**Runtime.** Figures 2 and 3 show that both FW-TSW and FW-TSW$^*$ scale linearly with the number $n$ and $d$. This is consistent with our complexity analysis.

**Memory scaling.** Figures 2 and 3 present the memory usage of FW-TSW and FW-TSW$^*$. All methods exhibit linear scaling with both the number of supports $n$ and the number of dimension $d$, consistent with the theoretical complexity analysis.

**Weiszfeld's Algorithm.** We empirically determined that setting the maximum iterations for Weiszfeld's algorithm to $T = 100$ provides substantial performance gains. This value is used as the default in our experiments unless otherwise noted.

## D.2 GRADIENT FLOW

**Gradient Flow on Point Cloud.** We perform a point cloud interpolation experiment to evaluate our method against two baseline approaches: SW (Bonneel et al., 2015) and Db-TSW (Tran et al., 2025a). Both the source and target point clouds are sampled from the ShapeNet Core-55 dataset (Chang et al., 2015), as illustrated in Figure 5. Following the experimental setup in (Nguyen et al., 2023), we use gradient approximation techniques to conduct Euler integration over 500 iterations with a step size of 0.01. Table 5 reports the Wasserstein distances between the interpolated point cloud and the target shape at iterations 100, 200, 300, 400, and 500, averaged over 5 runs. All methods use 100 projections. For TSW variants, we use 25 trees and 4 lines. Results are shown in Table 5.

**Gradient Flow on Images.** The experiment on synthetic MNIST-like images aims to learn a mapping from a noise distribution to a target distribution of 16 ordered digits, concurrently learning both the image content and their correct sequence.

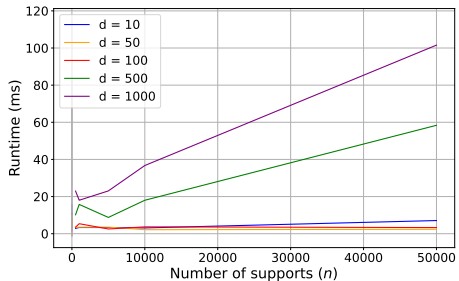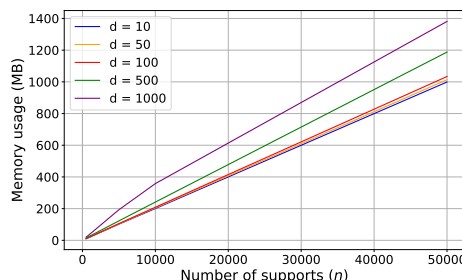

Figure 3: Execution time and memory usage of FW-TSW$^*$.

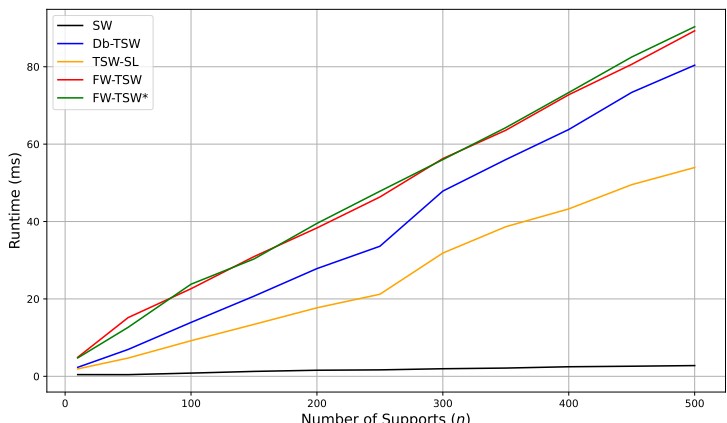

Figure 4: Runtime Comparison of FW-TSW and other methods.

The target distribution consists of 16 unique samples. Each sample represents a grayscale image of a digit (0 through 15). These are synthetic $28 \times 28$ images, flattened to 784-dimensional vectors. A scalar positional encoding, $i/16$ (where $i$ is the sample's index from 0 to 15), is appended to each flattened image vector. This results in each target sample $\nu_i$ being a 785-dimensional vector ($28 \times 28 + 1$). The set of these 16 vectors forms the target distribution $\nu$. The initial source distribution $\mu_0$ also comprises 16 samples, each initialized as a 785-dimensional vector from Gaussian noise. The discrepancy between the evolving source distribution $\mu_t$ and the target distribution $\nu$ is minimized using the Adam optimizer with a learning rate of $1 \times 10^{-3}$ applied to all methods.

For all Tree-Sliced Wasserstein (TSW) variants, including our proposed FW-TSW-DD and FW-TSW$^*$-DD, the number of sampled trees (for TSW variants, $L$) is set to 250 and the number of lines is set to $k = 4$ per tree.

To evaluate performance, the 16 samples of the current source distribution $\mu_t$ are first sorted based on their learned positional encoding values (the last dimension of each 785-dimensional sample). After sorting, the pairwise $L_2$ distance is computed between the image-only part (the first 784 dimensions) of these sorted reconstructed samples and the corresponding ordered ground truth target images.

To account for variability and the potential bi-modal nature of the $L_2$ metric (due to ordering success or failure), each experimental setup for each method is repeated 100 times with different random seeds. The reported $L_2$ values in Table 6 are percentiles (e.g., P25, P50/Median, P75) derived from these 100 runs.

Table 6 shows that at epoch 3000, FW-TSW and FW-TSW$^*$ consistently achieve lower $L_2$ values compared to Db-TSW. For instance, the median $L_2$ for FW-TSW and FW-TSW$^*$ is $0.50$ and $1.60$, respectively, while Db-TSW yields a median $L_2$ of $5.67$. Figure 6 visually substantiates this, illus-

| Methods | Step 100 | Step 200 | Step 300 | Step 400 | Step 500 |
|---------|----------|----------|----------|----------|----------|
| SW | **2.07e-04** | 1.39e-04 | 1.31e-04 | 1.35e-04 | 1.28e-04 |
| Db-TSW | 4.63e-04 | 7.54e-05 | 3.05e-05 | 1.90e-05 | 1.37e-05 |
| FW-TSW | 4.56e-04 | **7.17e-05** | 2.90e-05 | 1.67e-05 | 1.30e-05 |
| FW-TSW* | 4.93e-04 | 7.23e-05 | **2.74e-05** | **1.65e-05** | **1.15e-05** |

Table 5: Comparison of methods across training steps.

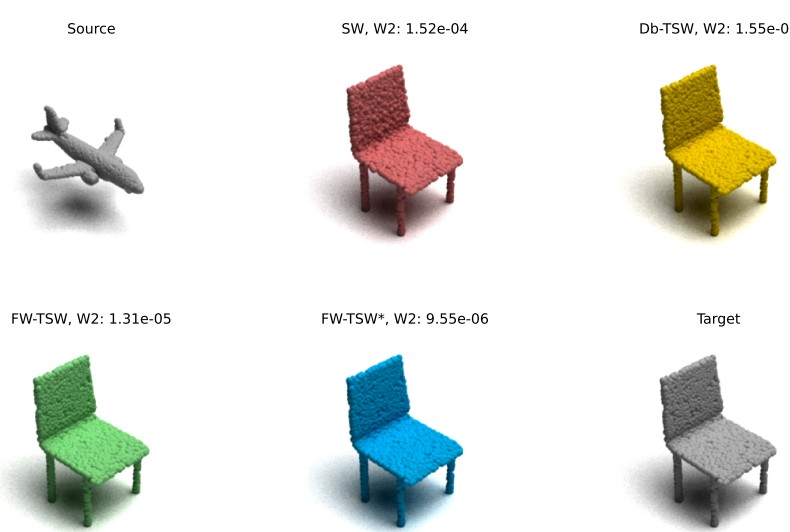

Figure 5: Point-cloud interpolation with Wasserstein distance at step 500.

trating that FW-TSW and FW-TSW* successfully learn both the image content and their ordering by epoch 3000. In contrast, Db-TSW still results in blurry images and incorrect ordering.

**Gradient Flow on synthetic data.** Table 1 presents the performance of our proposed methods alongside various baselines on the 25 Gaussians dataset. The low standard deviation observed for FW-TSW and FW-TSW* highlights their stable and consistent convergence behavior. For the Tree-Sliced Wasserstein (TSW) and its variants, we employ $L = 25$ trees and $k = 4$ lines. We set $L = 100$ projections for other sliced methods. All models are trained for 2500 steps using the Adam optimizer. The source and target distributions contain 500 samples each.

To ensure a fair comparison, we perform an ablation study over a range of learning rates for each method. The results reported in the main table correspond to the best-performing learning rate for each method. Complete results of the ablation study are shown in Table 7.

### D.3 TOPIC MODELING

In this section, we present the details of our Topic Modeling experiments.

**Topic Modeling.** Topic modeling (Blei et al., 2003) is a long-standing task in Natural Language Processing that aims to discover latent thematic structures within document corpora. Typically, documents $\mathbf{x}$ are represented using a bag-of-words model, while the topic proportions $\theta$ are modeled as a discrete distribution over topics. Recent advances utilize variational autoencoder (VAE) to address this task, where an encoder network $\varphi$ estimates the posterior distribution $q_\varphi(\theta|\mathbf{x})$, and a decoder network $\psi$ reconstructs documents as $\hat{\mathbf{x}} = \psi(\theta)$. The objective function for training such

Table 6: $L_2$ distances at various percentiles (e.g., P25, P50/Median, P75, where 'P' denotes percentile) comparing Db-TSW with our proposed FW-TSW and FW-TSW* on the gradient flow task for synthetic MNIST-like images, evaluated at selected training timesteps. These $L_2$ values represent the distance computed between pixel data of source (reconstructed) and target (ground truth) images, after both have been sorted using positional encodings. Lower $L_2$ indicate better performance. For the Step 3000 metrics, our methods demonstrate superior performance.

| Methods | Steps | | | | | | |
| --- | --- | --- | --- | --- | --- | --- | --- |
| | 1000 | 2000 | 2500 | 3000 | | | 4000 |
| | $L_2$ (P50) | $L_2$ (P50) | $L_2$ (P50) | $L_2$ (P25) | $L_2$ (P50) | $L_2$ (P75) | $L_2$ (P50) |
| Db-TSW | 12.92 | 10.37 | 10.12 | 1.27 | 5.67 | 6.58 | 0.05 |
| FW-TSW (Ours) | 12.91 | 10.26 | 10.01 | **0.26** | **0.50** | **4.13** | 0.05 |
| FW-TSW* (Ours) | 12.93 | 10.30 | 10.09 | 0.88 | 1.60 | 5.12 | 0.05 |

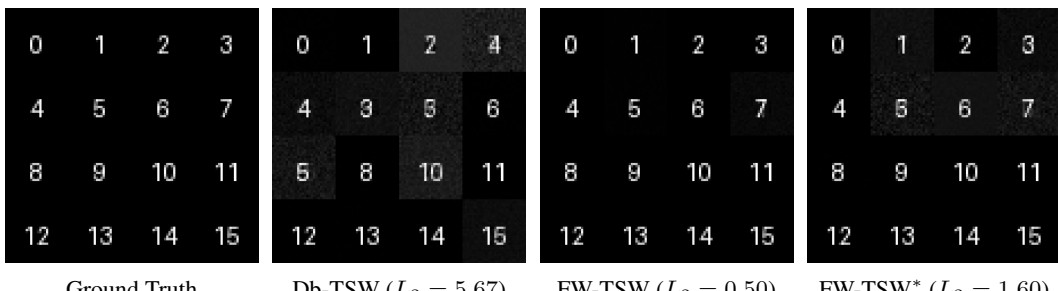

| Ground Truth | Db-TSW ($L_2 = 5.67$) | FW-TSW ($L_2 = 0.50$) | FW-TSW* ($L_2 = 1.60$) |

Figure 6: Image reconstruction and ordering by gradient flow methods on synthetic MNIST-like digits (epoch 3000). Ground Truth (far left): numbers 0–15 ordered left-to-right, top-to-bottom. Other panels show reconstructions reflecting each method's median $L_2$ performance. Our proposed FW-TSW and FW-TSW* produce correctly ordered images, unlike Db-TSW's misordered results.

models is usually given by

$$\mathcal{L} = \mathbb{E}_{p(\mathbf{x})q(\theta|\mathbf{x})}\big[\mathrm{CE}(\mathbf{x}, \hat{\mathbf{x}})\big] + \lambda \, \mathrm{KL}(q(\theta|\mathbf{x})\|p(\theta)),$$

where $\mathrm{CE}(\cdot, \cdot)$ is the cross-entropy reconstruction loss and $\mathrm{KL}(\cdot\|\cdot)$ is the Kullback–Leibler divergence regularizing the posterior to match the prior $p(\theta)$.

In our experiment, we replace the KL divergence term by our FW-TSW and FW*-TSW. We benchmark these against other sliced Wasserstein methods in Euclidean setting (Nguyen et al., 2024b; Tran et al., 2025a; 2024e; Bonneel et al., 2015), as well as classical topic modeling approaches such as Latent Dirichlet Allocation (LDA) (Blei et al., 2003), ProdLDA (Srivastava & Sutton, 2017), and Wasserstein Topic Model (WTM) (Nan et al., 2019).

**Datasets.** We evaluate our methods on three widely used benchmark datasets for topic modeling:

- **M10** (Pan et al., 2016): A subset of the CiteSeer$^X$ digital library, consisting of over 8,000 academic documents across 10 research topics.

- **DBLP** (Pan et al., 2016): A bibliographic dataset in computer science, containing more than 50,000 documents from 4 research domains.

- **BBC** (Greene & Cunningham, 2006): A collection of over 2,000 news articles published by the BBC, covering 5 topical categories.

For preprocessing, we convert all text to lowercase, remove punctuation, perform lemmatization, filter out short words (fewer than 3 characters), and discard short documents (fewer than 3 words). Detailed statistics of the preprocessed datasets are reported in Table 8.

**Metrics.** A common approach to evaluating topic models involves assessing two key aspects: topic coherence and topic diversity. We adopt the $C_V$ (CV) measure ↑, which has been demonstrated to strongly align with human judgment (Röder et al., 2015), as our primary coherence metric. For topic diversity, we use the IRBO metric ↑ (Terragni et al., 2021), a widely accepted measure capturing the

Table 7: Average Wasserstein distance between source and target distributions over 5 runs on the 25 Gaussians dataset with different learning rate $\eta = \{0.001, 0.005, 0.01, 0.05, 0.1\}$. All methods use 100 projecting directions.

| Method | $\eta$ | Iteration | | | | |
|---|---|---|---|---|---|---|
| | | 500 | 1000 | 1500 | 2000 | 2500 |
| SW | 0.001 | 4.20e-01 ± 6.2e-03 | 1.53e-01 ± 2.3e-03 | 7.83e-02 ± 2.3e-03 | 5.04e-02 ± 1.9e-03 | 3.61e-02 ± 1.4e-03 |
| | 0.005 | 4.02e-02 ± 3.0e-03 | 1.82e-02 ± 1.8e-03 | 1.08e-02 ± 1.3e-03 | 7.91e-03 ± 9.2e-04 | 6.53e-03 ± 1.2e-03 |
| | 0.01 | 1.92e-02 ± 2.0e-03 | 8.68e-03 ± 1.5e-03 | 6.83e-03 ± 9.2e-04 | 5.93e-03 ± 1.5e-03 | 5.51e-03 ± 1.4e-03 |
| | 0.05 | 6.16e-03 ± 7.3e-04 | 4.98e-03 ± 7.8e-04 | 4.48e-03 ± 1.1e-03 | 3.99e-03 ± 1.4e-03 | 3.47e-03 ± 1.5e-03 |
| | 0.1 | 3.65e-03 ± 1.3e-03 | 2.42e-03 ± 8.0e-04 | 2.13e-03 ± 9.0e-04 | 1.69e-03 ± 9.8e-04 | 1.01e-03 ± 9.5e-04 |
| SWGG | 0.001 | 6.48e-01 ± 2.6e-02 | 3.49e-01 ± 4.4e-02 | 1.73e-01 ± 5.9e-02 | 6.77e-02 ± 4.3e-02 | 2.32e-02 ± 2.1e-02 |
| | 0.005 | 2.06e-02 ± 1.6e-02 | 8.26e-04 ± 1.6e-03 | 4.50e-06 ± 4.2e-06 | 5.29e-06 ± 8.6e-06 | 3.42e-06 ± 5.5e-06 |
| | 0.01 | 7.67e-04 ± 1.4e-03 | 4.85e-06 ± 5.5e-06 | 2.91e-06 ± 2.4e-06 | 2.72e-06 ± 5.3e-06 | 2.91e-06 ± 5.7e-06 |
| | 0.05 | 2.26e-05 ± 8.5e-06 | 2.93e-05 ± 1.3e-05 | 4.61e-05 ± 1.7e-05 | 5.81e-05 ± 2.2e-05 | 9.15e-05 ± 2.5e-05 |
| | 0.1 | 5.39e-05 ± 1.2e-05 | 7.03e-05 ± 2.4e-05 | 1.38e-04 ± 1.6e-05 | 1.98e-04 ± 3.4e-05 | 5.33e-03 ± 8.8e-03 |
| LCVSW | 0.001 | 3.47e-01 ± 5.0e-03 | 6.99e-02 ± 2.5e-03 | 2.35e-02 ± 2.0e-03 | 1.25e-02 ± 1.7e-03 | 9.04e-03 ± 1.4e-03 |
| | 0.005 | 8.07e-03 ± 1.5e-03 | 5.02e-03 ± 1.0e-03 | 4.45e-03 ± 9.1e-04 | 4.08e-03 ± 9.2e-04 | 4.17e-03 ± 9.2e-04 |
| | 0.01 | 4.15e-03 ± 1.4e-03 | 3.77e-03 ± 9.3e-04 | 3.75e-03 ± 9.0e-04 | 3.83e-03 ± 1.0e-03 | 3.81e-03 ± 9.2e-04 |
| | 0.05 | 2.33e-03 ± 1.6e-03 | 2.11e-03 ± 1.3e-03 | 2.18e-03 ± 1.4e-03 | 2.13e-03 ± 1.3e-03 | 2.36e-03 ± 1.4e-03 |
| | 0.1 | 1.54e-03 ± 1.1e-03 | 1.40e-03 ± 8.7e-04 | 7.84e-04 ± 5.6e-04 | 5.73e-04 ± 6.3e-04 | 6.84e-04 ± 7.9e-04 |
| TSW-SL | 0.001 | 3.50e-01 ± 4.8e-03 | 8.12e-02 ± 2.5e-03 | 1.09e-02 ± 9.0e-04 | 2.73e-03 ± 8.9e-04 | 2.36e-04 ± 3.3e-04 |
| | 0.005 | 1.12e-03 ± 9.7e-04 | 1.37e-06 ± 8.7e-08 | 1.07e-06 ± 4.8e-08 | 9.13e-07 ± 5.2e-08 | 8.76e-07 ± 1.1e-07 |
| | 0.01 | 7.73e-06 ± 7.4e-07 | 4.73e-06 ± 2.0e-07 | 4.49e-06 ± 2.6e-07 | 4.11e-06 ± 1.4e-07 | 3.76e-06 ± 3.7e-07 |
| | 0.05 | 9.25e-05 ± 4.5e-06 | 8.72e-05 ± 4.8e-06 | 9.03e-05 ± 2.1e-06 | 8.70e-05 ± 4.1e-06 | 8.76e-05 ± 9.2e-06 |
| | 0.1 | 3.87e-04 ± 4.0e-05 | 3.20e-04 ± 3.8e-05 | 3.53e-04 ± 5.4e-05 | 3.29e-04 ± 3.3e-05 | 3.52e-04 ± 2.0e-05 |
| Db-TSW | 0.001 | 3.68e-01 ± 5.0e-03 | 1.06e-01 ± 3.4e-03 | 1.91e-02 ± 1.1e-03 | 4.14e-03 ± 9.0e-04 | 3.44e-04 ± 6.7e-04 |
| | 0.005 | 3.42e-03 ± 7.9e-04 | 1.55e-06 ± 1.2e-07 | 1.10e-06 ± 9.2e-08 | 9.50e-07 ± 6.1e-08 | 8.55e-07 ± 5.6e-08 |
| | 0.01 | 8.49e-06 ± 5.5e-07 | 5.52e-06 ± 1.7e-07 | 4.90e-06 ± 3.6e-07 | 4.50e-06 ± 2.5e-07 | 4.28e-06 ± 3.3e-07 |
| | 0.05 | 9.98e-05 ± 7.3e-06 | 9.75e-05 ± 5.1e-06 | 9.54e-05 ± 6.0e-06 | 1.00e-04 ± 1.1e-05 | 9.82e-05 ± 9.3e-06 |
| | 0.1 | 4.63e-04 ± 4.5e-05 | 4.00e-04 ± 3.9e-05 | 3.50e-04 ± 2.9e-05 | 3.69e-04 ± 1.9e-05 | 3.63e-04 ± 3.6e-05 |
| Db-TSW$^\perp$ | 0.001 | 3.72e-01 ± 4.6e-03 | 1.06e-01 ± 4.0e-03 | 1.88e-02 ± 9.9e-04 | 3.40e-03 ± 6.5e-04 | 3.61e-04 ± 5.4e-04 |
| | 0.005 | 2.70e-03 ± 9.0e-04 | 1.79e-06 ± 2.0e-07 | 1.25e-06 ± 9.7e-08 | 1.14e-06 ± 5.6e-08 | 1.03e-06 ± 4.8e-08 |
| | 0.01 | 1.36e-05 ± 1.0e-06 | 7.95e-06 ± 7.3e-07 | 6.89e-06 ± 6.5e-07 | 6.20e-06 ± 3.1e-07 | 6.89e-06 ± 7.0e-07 |
| | 0.05 | 1.22e-04 ± 7.1e-06 | 1.13e-04 ± 5.2e-06 | 1.11e-04 ± 8.3e-06 | 1.16e-04 ± 8.3e-06 | 1.22e-04 ± 1.3e-05 |
| | 0.1 | 4.44e-04 ± 4.7e-05 | 4.44e-04 ± 4.0e-05 | 4.53e-04 ± 7.1e-05 | 4.00e-04 ± 4.9e-05 | 4.33e-04 ± 8.1e-05 |
| FW-TSW | 0.001 | 3.68e-01 ± 4.9e-03 | 1.06e-01 ± 3.6e-03 | 1.96e-02 ± 8.4e-04 | 5.09e-03 ± 6.5e-04 | 6.98e-04 ± 5.0e-04 |
| | 0.005 | 2.40e-03 ± 8.9e-04 | 1.51e-06 ± 1.4e-07 | 1.03e-06 ± 1.0e-07 | 9.18e-07 ± 4.1e-08 | 8.40e-07 ± 2.6e-08 |
| | 0.01 | 8.66e-06 ± 6.0e-07 | 5.40e-06 ± 1.8e-07 | 4.84e-06 ± 4.0e-07 | 4.55e-06 ± 2.3e-07 | 4.33e-06 ± 3.3e-07 |
| | 0.05 | 1.03e-04 ± 1.0e-05 | 9.67e-05 ± 5.6e-06 | 9.93e-05 ± 6.9e-06 | 1.01e-04 ± 8.0e-06 | 9.70e-05 ± 1.1e-05 |
| | 0.1 | 3.77e-04 ± 4.0e-05 | 3.68e-04 ± 4.6e-05 | 3.57e-04 ± 4.9e-05 | 3.72e-04 ± 3.6e-05 | 4.04e-04 ± 2.9e-05 |
| FW-TSW$^*$ | 0.001 | 3.68e-01 ± 5.1e-03 | 1.06e-01 ± 4.2e-03 | 1.90e-02 ± 1.6e-03 | 4.06e-03 ± 1.3e-03 | 1.11e-03 ± 1.6e-03 |
| | 0.005 | 2.59e-03 ± 9.3e-04 | 1.50e-06 ± 8.9e-08 | 1.11e-06 ± 6.6e-08 | 9.04e-07 ± 1.1e-07 | 8.29e-07 ± 4.7e-08 |
| | 0.01 | 8.66e-06 ± 7.1e-07 | 5.79e-06 ± 1.9e-07 | 4.83e-06 ± 4.2e-07 | 4.27e-06 ± 2.7e-07 | 4.10e-06 ± 1.5e-07 |
| | 0.05 | 9.80e-05 ± 2.5e-06 | 9.50e-05 ± 3.1e-06 | 1.03e-04 ± 1.1e-05 | 9.09e-05 ± 2.2e-06 | 9.73e-05 ± 6.8e-06 |
| | 0.1 | 4.29e-04 ± 5.9e-05 | 3.53e-04 ± 2.6e-05 | 3.89e-04 ± 2.9e-05 | 3.74e-04 ± 5.2e-05 | 3.80e-04 ± 2.3e-05 |

Table 8: Dataset statistics and hyperparameters.

| | Dataset statistics | | | Hyperparameters | | |
|---|---|---|---|---|---|---|
| Dataset | #Docs | #Labels | #Words | #Projections | Batch size | Dropout rate |
| DBLP | 54595 | 4 | 1513 | 1000 | 512 | 0.2 |
| M10 | 8355 | 10 | 1696 | 2000 | 64 | 0.5 |
| BBC | 2225 | 5 | 2949 | 8000 | 256 | 0.05 |

distinctness among topics. Intuitively, topic coherence quantifies how frequently the top words of a topic co-occur within the same documents across the corpus, while topic diversity reflects the degree to which the topics are well-separated and capture different themes. We report the mean and standard deviation across 10 independent runs.

**Training.** We employ OCTIS (Terragni et al., 2021), a widely adopted framework for training and evaluating topic models. We adhere to the experimental settings outlined in (Adhya & Sanyal, 2025), employing a Euclidean latent space with a Dirichlet prior. Each model is trained for 100 epochs. The weighting hyperparameter is systematically varied over the interval $[0.5, 10]$ in increments of

Table 9: Topic diversity by IRBO ($\uparrow$) on DBLP, M10, and BBC.

| Method | DBLP | M10 | BBC |
|---|---|---|---|
| LDA (Blei et al., 2003) | $0.837_{\pm 0.023}$ | $0.915_{\pm 0.012}$ | $0.917_{\pm 0.016}$ |
| ProdLDA (Srivastava & Sutton, 2017) | $1.000_{\pm 0.000}$ | $0.997_{\pm 0.002}$ | $1.000_{\pm 0.000}$ |
| WTM (Nan et al., 2019) | $0.953_{\pm 0.035}$ | $0.812_{\pm 0.060}$ | $0.997_{\pm 0.003}$ |
| SW-TM (Bonneel et al., 2015) | $0.984_{\pm 0.016}$ | $0.971_{\pm 0.008}$ | $0.998_{\pm 0.004}$ |
| RPSW-TM (Nguyen et al., 2024b) | $0.992_{\pm 0.007}$ | $0.969_{\pm 0.010}$ | $0.994_{\pm 0.010}$ |
| EBRPSW-TM (Nguyen et al., 2024b) | $0.991_{\pm 0.007}$ | $0.963_{\pm 0.013}$ | $0.999_{\pm 0.003}$ |
| TSW-SL-TM (Tran et al., 2024e) | $0.989_{\pm 0.014}$ | $0.981_{\pm 0.005}$ | $0.999_{\pm 0.003}$ |
| Db-TSW-TM (Tran et al., 2025a) | $0.988_{\pm 0.012}$ | $0.805_{\pm 0.044}$ | $1.000_{\pm 0.000}$ |
| FW-TSW-TM (ours) | $0.944_{\pm 0.034}$ | $0.812_{\pm 0.089}$ | $0.998_{\pm 0.004}$ |
| FW*-TSW-TM (ours) | $0.961_{\pm 0.033}$ | $0.913_{\pm 0.036}$ | $0.998_{\pm 0.003}$ |

0.5. For tree-based approaches, the number of trees is fixed at 100. All other training parameters are detailed in Table 8.

**Topic Diversity.** We provide topic diversity result in Table 9.

### D.4 DIFFUSION MODELS

**Diffusion Models.** A prominent category of generative models, diffusion models (Sohl-Dickstein et al., 2015; Ho et al., 2020), are recognized for their capacity to synthesize high-fidelity data. This section outlines their fundamental mechanisms and sets the stage for the enhancements introduced by our work. The core idea involves a forward process where an initial data sample $q(x_0)$ is systematically degraded by the incremental addition of Gaussian noise across $T$ discrete timesteps. This transformation is formally described by:

$$q(x_{1:T}|x_0) = \prod_{t=1}^{T} q(x_t|x_{t-1}),$$

with each step $q(x_t|x_{t-1})$ in this sequence being a Gaussian transition defined as:

$$q(x_t|x_{t-1}) = \mathcal{N}(x_t; \sqrt{1-\beta_t}x_{t-1}, \beta_t I).$$

The parameters $\beta_t$ in this equation correspond to a predetermined noise variance schedule.

Conversely, the generative aspect of these models lies in learning the reverse process: to denoise a corrupted sample and recover the original data structure. This involves parameterizing the reverse transitions $p_\theta(x_{t-1}|x_t)$ using a neural network with parameters $\theta$. The complete reverse process is given by:

$$p_\theta(x_{0:T}) = p(x_T) \prod_{t=1}^{T} p_\theta(x_{t-1}|x_t),$$

where each individual reverse step is also modeled as a Gaussian:

$$p_\theta(x_{t-1}|x_t) = \mathcal{N}(x_{t-1}; \mu_\theta(x_t, t), \sigma_t^2 I).$$

The training objective is generally to maximize the Evidence Lower Bound (ELBO). This is tantamount to minimizing the Kullback-Leibler (KL) divergence between the true, but intractable, posterior $q(x_{t-1}|x_t)$ and the model's learned approximation $p_\theta(x_{t-1}|x_t)$, summed over all timesteps:

$$L = -\sum_{t=1}^{T} \mathbb{E}_{q(x_t)} \left[ \text{KL}(q(x_{t-1}|x_t)||p_\theta(x_{t-1}|x_t)) \right] + C,$$

where $C$ denotes a constant term and $\text{KL}(\cdot||\cdot)$ is the KL divergence.

**Denoising Diffusion GANs.** A primary limitation of standard diffusion models is their considerable sampling latency, which can hinder their use in time-sensitive applications. Denoising Diffusion GANs (DDGANs) (Xiao et al., 2021) were developed to mitigate this inefficiency. DDGANs

reframe each denoising step as a conditional generation task handled by a multimodal Generative Adversarial Network (GAN). This architecture permits larger individual denoising steps, drastically cutting down the total number of required steps to as few as four. Consequently, DDGANs can achieve sampling speeds over 2000 times faster than their traditional counterparts, without a significant compromise in the quality or diversity of the generated samples. The implicit denoising distribution within DDGANs is expressed as:

$$p_\theta(x_{t-1}|x_t) = \int p_\theta(x_{t-1}|x_t, \epsilon) G_\theta(x_t, \epsilon) d\epsilon, \quad \epsilon \sim \mathcal{N}(0, I).$$

Originally, Xiao et al. (2021) trained the model parameters $\theta$ via an adversarial objective:

$$\min_\phi \sum_{t=1}^{T} \mathbb{E}_{q(x_t)}[D_{adv}(q(x_{t-1}|x_t)||p_\phi(x_{t-1}|x_t))],$$

with $D_{adv}$ signifying the adversarial loss. However, Nguyen et al. (2024b) proposed an alternative by substituting this adversarial loss with the Augmented Generalized Mini-batch Energy (AGME) distance. For two distributions, $\mu$ and $\nu$, and a mini-batch size $n \geq 1$, the AGME, when employing a Sliced Wasserstein (SW) kernel and a nonlinear function $g : \mathbb{R}^d \to \mathbb{R}$ to define $f(x) = (x, g(x))$, is given by:

$$\text{AGME}_b^2(\mu, \nu; g) = \text{GME}_b^2(\tilde{\mu}, \tilde{\nu}),$$

where $\tilde{\mu} = f_\sharp \mu$ and $\tilde{\nu} = f_\sharp \nu$. The underlying Generalized Mini-batch Energy (GME) distance (Salimans et al., 2018) is formulated as:

$$\text{GME}_b^2(\mu, \nu) = 2\mathbb{E}[D(P_X, P_Y)] - \mathbb{E}[D(P_X, P'_X)] - \mathbb{E}[D(P_Y, P'_Y)],$$

where $X, X' \overset{i.i.d.}{\sim} \mu^{\otimes m}$ and $Y, Y' \overset{i.i.d.}{\sim} \nu^{\otimes m}$. The empirical distributions $P_X$ and $P_Y$ are constructed from mini-batches, e.g., $P_X = \frac{1}{m} \sum_{i=1}^{m} \delta_{x_i}$ for $X = (x_1, \ldots, x_m)$. The metric $D$ in the GME formulation can be any valid distance. In this work, we explore the use of Sliced Wasserstein (SW) variants and our proposed Tree-Sliced Wasserstein (TSW) variants as choices for $D$.

**Setting.** Our experimental configuration largely mirrors that of Nguyen et al. (2024b) and Tran et al. (2025a) in terms of model architecture and foundational hyperparameters. All models are trained for 1800 epochs. For Tree-Sliced methodologies, including our novel techniques, we configure $L = 2500$ sampled trees and $k = 4$ lines per tree, following Tran et al. (2025a). In contrast, for vanilla SW and its associated variants, $L = 10000$ projections are used, consistent with Nguyen et al. (2024b). Learning rates are also adopted from Nguyen et al. (2024b), specifically $lr_d = 1.25 \times 10^{-4}$ and $lr_g = 1.6 \times 10^{-4}$. For our FW-TSW* method, we used a $\kappa$ scheduling scheme as in Nguyen et al. (2024b). The standard deviation for tree sampling is $0.1$, as per Tran et al. (2025a). Runtime evaluations are conducted using a batch size of 128 on two NVIDIA H100 GPUs. Our results for FW-TSW and FW-TSW* are averaged over 10 runs while other results are obtained from previous results.

### D.5  HARDWARE SETTINGS

All experiments utilized an Intel Xeon Platinum 8580 CPU. Gradient flow experiments were performed on a single NVIDIA H100 GPU, while denoising diffusion experiments were executed in parallel across two NVIDIA H100 GPUs.

## E  BROADER IMPACTS

The FW-TSW method introduced in this paper has significant societal implications by improving the accuracy and flexibility of optimal transport techniques across a wide range of real-world applications. It has the potential to advance fields such as healthcare—where improved image processing can support more precise medical diagnostics—and the arts and entertainment industry, by enabling more refined and creative generative models. Additionally, its ability to operate effectively in dynamic environments unlocks new opportunities for real-time data analysis and informed decision-making in domains like finance, logistics, and environmental monitoring. In essence, FW-TSW enhances the practicality and reach of modern computational tools, promoting innovation and contributing to overall societal advancement.

