# OpenReview forum: "Revisiting Tree-Sliced Wasserstein Distance Through the Lens of the Fermat–Weber Problem"
_ICLR.cc/2026/Conference — ICLR 2026 Poster_

### Official Review · Reviewer_jHMb · 2025-10-26

**Soundness:** 3
**Presentation:** 2
**Contribution:** 2
**Rating:** 4
**Confidence:** 5

**Summary:**

While Optimal Transport (OT) allows to compare data in a 'meaningful' way, it is computionally costly. Slicing strategies allow for the definition of more efficient OT-based metrics.
In particular, this paper introduces a new variant of the Tree-Sliced Wasserstein (TSW) distance, called the Fermat-Weber Tree-Sliced Wasserstein (FW-TSW). It is presented as an efficient alternative to standard Sliced Wasserstein (SW), utilizing tree-structured metric spaces to better capture data geometry while maintaining low computational cost. The key insight leveraged is that TSW incorporates both directional and positional information distinguishing it from the purely directional SW. A key question for TSW is the generation of random trees. The authors propose FW-TSW, which explicitly integrates the geometric median (based on the Fermat-Weber problem) into the tree construction process to enhance this positional sensitivity. They claim that this leads to further performance improvements over TSW, which is empirically validated in experiments involving diffusion model training and gradient flows.

**Strengths:**

Novelty: Sampling strategies for sliced Wasserstein and kernel distances are an active field. Beyond projections onto the line, it is valuable to study sampling strategies for TSW, as the combined use of directional and positional information has been relatively underexplored.

Structural Design: The integration of the geometric median into the tree construction mechanism is an intuitive apprach for sampling meaningful tree projections.

Theoretical Contribution: Since there are limited theoretical results for TSW, the bound presented in Theorem 4.5 is an interesting result that provides new theoretical insight into the properties of Tree-Sliced distances.

Empirical Validation: The experiments are targeted at typical applications of sliced OT and provide good support for the proposed FW-TSW method. There are extensive ablation studies in the supplementary material.

**Weaknesses:**

Limited Impact: While the question for 'smart' tree sampling is interesting and deserves further studiy, the contribution seems rather incremental.

Theoretical Complexity: Most of the theoretical results are assessed are  straightforward with the exception of  Theorem 4.5.

Doubts about empirical validation: Much of the experiments focus on comparing SW and TSW for a predefined number of projections. However, SW is faster, so it might be fairer to compare the two for a predefined run time. After all, the number of projections is very important for the Monte Carlo estimates.

Bound Limitation: The utility of the Theorem 4.5 bound is questionable due to a potentially large additive constant, especially for a high number of splits $k$.

Applicability: The practical relevance of the demonstrated applications is limited outside of a dedicated Computational Optimal Transport research area. To my knowledge, TSW is not widely used for practical application (unlike standard OT or SW).

Clarity and Notation: Section 2.2 suffers from inconsistent and confusing notation. Key terms like $\mathbb T_k^d$ are undefined, and the subscript $l$ in Equation (4) is unexplained, hindering comprehension. The rather central closed-form Wasserstein formula for trees is not provided in the paper, but only in the references. It is also confusing that the supplementary material around (24) uses partly different notation.

References: Relevant references on sampling strategies for sliced divergences, e.g. (Hertrich et al., 2025) and (Sisouk et al., 2025) are missing.

**Questions:**

1. Scaling in (14): In Equation (14), why is the identity matrix $\mathbf{I}_d$ not scaled according to the data's covariance or variance?

2. Subscripts in (15): The measure dependence should be made explicit: the subscripts $\mu, \nu$ of $\Sigma$ should already appear in Equation (15).

3. Choice of Central Point: Why was the geometric median chosen over simply using the barycenter of the measures in the tree construction? Is it because of Wasserstein-1 instead of Wasserstein-2? Did you try using the barycenter?

4. $k$-Disjoint Union: In Equation (6), what is the precise definition and calculation of the distance on the $k$-times disjoint union of $\mathbb R$?

5. Typos and Formatting: Please correct the formatting error "Equation equation 4" (bottom of p. 6 and elsewhere). Ensure distinct notation is used for the exact distance (6) versus its approximation (8). Also, correct Equation (28) by removing the spurious subscript $l$ on the left-hand side.

---

> ### Author Response · Authors · 2025-11-27
> **Response [1/3]**
>
> We are grateful for the Reviewer’s feedback. To ensure a structured response, we have addressed related questions and concerns together. Please find the revised manuscript with all modifications highlighted in blue.
>
> ---
>
> **W1. Limited Impact: While the question for 'smart' tree sampling is interesting and deserves further studiy, the contribution seems rather incremental.**
>
> **W5. Applicability: The practical relevance of the demonstrated applications is limited outside of a dedicated Computational Optimal Transport research area. To my knowledge, TSW is not widely used for practical application (unlike standard OT or SW).**
>
> **Answer W1+W5.** While the original Sliced Wasserstein (SW) distance was introduced long ago, improving its core components, such as slice-sampling strategies, projection mechanisms, and data-dependent versus data-independent sampling, remains an active line of research, with many works reporting significant empirical gains.
>
> In this work, we advance the Tree-Sliced Wasserstein (TSW) framework by proposing a new sampling scheme for tree slices that incorporates a geometric-mean based sampling of the tree root. Within the context of TSW, this constitutes a novel idea. Moreover, unlike in SW variants, there is no direct analogue of this approach: the presence of a root in tree-metric systems is unique to TSW, encoding positional structure of the slice, whereas SW relies solely on directional information.
>
> Although TSW is relatively new compared to classical OT and SW, empirical evidence highlights its strong potential. For instance, in our largest experiment on denoising diffusion models, the best-performing SW variant achieves a FID of $2.70$, while TSW-based methods perform substantially better. Our proposed FW-TSW (root sampling only) attains a FID of $2.336$, and FW-TSW$^\ast$ (root + directional sampling) further improves this slightly to $2.315$. These results underscore both the promise of the TSW framework and the value in exploring its new geometric components.
>
> **W2 Theoretical Complexity: Most of the theoretical results are assessed are straightforward with the exception of Theorem 4.5.**
>
> **Answer W2.** These theoretical results are included both for completeness and to provide formal guarantees for TSW. We acknowledge that some of the derivations may appear straightforward to readers with a strong background in Optimal Transport. However, given the broader audience of our work, we believe presenting these results explicitly is necessary.
>
> **W3. Doubts about empirical validation: Much of the experiments focus on comparing SW and TSW for a predefined number of projections. However, SW is faster, so it might be fairer to compare the two for a predefined run time. After all, the number of projections is very important for the Monte Carlo estimates.**
>
> **Answer W3.** We thank the reviewer for the suggestion; indeed, comparing methods under a predefined runtime is a practical perspective. However, beyond runtime, memory consumption is a critical bottleneck. We note that memory usage scales linearly with the number of projections for both SW and TSW, which limits the number of projections feasible for SW. For example, in our Diffusion Model experiments, scaling SW to a higher number of projections to match FW-TSW's runtime would exceed the memory capacity of our hardware. Furthermore, to the best of our knowledge, comparing baselines with a fixed number of projections is the standard protocol in prior Sliced Wasserstein literature, even when proposed methods introduce additional computational costs.

---

> > ### Author Response · Authors · 2025-11-27
> > **Response [2/3]**
> >
> > **W4. Bound Limitation: The utility of the Theorem 4.5 bound is questionable due to a potentially large additive constant, especially for a high number of splits $k$.**
> >
> > **Answer W4.** The bound in Theorem 4.5, although seemingly simple, represents the first attempt to relate TSW variants to a classical OT distance (in our case, the $W_2$ distance). The main challenge in establishing such a bound lies in the structural and metric complexity of tree systems. Unlike SW, where projections reduce the geometry to one-dimensional lines in Euclidean space, TSW operates on tree metrics with branching structures, roots, and piecewise geodesics. This richer geometry makes even basic inequalities substantially more delicate to derive.
> >
> > Regarding the bound in (23), although it contains an additive term of the form $k(k-1),\frac{2\pi^{d/2}}{\Gamma((d+1)/2)}\Gamma(1/2) f(v^*)$, the dimensional factor in front of $k(k-1)$ decays super-exponentially in $d$. Using Stirling’s approximation, one obtains $\frac{2\pi^{d/2}\Gamma(1/2)}{\Gamma((d+1)/2)} = \mathcal{O}((cd)^{-d/2})$ for some constant $c>0$, showing that this additive contribution rapidly becomes negligible in high dimensions.
> >
> > Our empirical results further support this observation: a relatively small and fixed number of splits (around $k \approx 10$) already achieves high accuracy, with additional splits offering only marginal improvements. Scaling $k$ aggressively with $d$ would not only be impractical but also ineffective. In high dimensions, random directions become nearly orthogonal and concentrate in a thin spherical shell, so extra projections add little new geometric information about the distributions. As a result, taking $k$ extremely large would introduce many redundant projections without improving the quality of the approximation.
> >
> > Thus, in realistic settings, $k$ is typically small and never approaches the super-exponential growth with respect to $d$ that would be necessary for the additive term to become significant. The bound therefore remains meaningful and informative across all practical regimes.
> >
> >
> >
> > **W6. Clarity and Notation: Section 2.2 suffers from inconsistent and confusing notation. Key terms like $\mathbb{T}_k^d$ are undefined, and the subscript $l$ in Equation (4) is unexplained, hindering comprehension. The rather central closed-form Wasserstein formula for trees is not provided in the paper, but only in the references. It is also confusing that the supplementary material around (24) uses partly different notation.**
> >
> > **Q2. Subscripts in (15): The measure dependence should be made explicit: the subscripts
> >  of $\mu$, $\nu$ and $\Sigma$ should already appear in Equation (15).**
> >
> > **Q5. Typos and Formatting: Please correct the formatting error "Equation equation 4" (bottom of p. 6 and elsewhere). Ensure distinct notation is used for the exact distance (6) versus its approximation (8). Also, correct Equation (28) by removing the spurious subscript $l$ on the left-hand side.**
> >
> > **Answer W6+Q2+Q5.** We thank the Reviewer for identifying these points, which help us further clarify the manuscript. We have incorporated the corresponding revisions into the updated version.
> >
> > **W7. References: Relevant references on sampling strategies for sliced divergences, e.g. (Hertrich et al., 2025) and (Sisouk et al., 2025) are missing.**
> >
> > **Answer W7.** We thank the Reviewer for bringing to our attention additional related work. However, we were unable to locate the first citation (Hertrich et al., 2025) and could only find the second reference (Sisouk et al., 2025):
> >
> > Keanu Siouk et al., A User’s Guide to Sampling Strategies for Sliced Optimal Transport. TMLR 06/2025
> >
> > This work concerns sampling strategies for Sliced OT, which is indeed relevant, and we appreciate the Reviewer pointing it out. We kindly ask the Reviewer to provide further details on the first reference so that we may properly examine it and include it in our literature review and manuscript if appropriate.

---

> > > ### Author Response · Authors · 2025-11-27
> > > **Response [3/3]**
> > >
> > > **Q1. Scaling in (14): In Equation (14), why is the identity matrix $\mathbf{I}_d$ not scaled according to the data's covariance or variance?**
> > >
> > > **Answer Q1.** In practice, since the data is typically normalized, we find that using a unit deviation yields stable behavior and performs well across most datasets.
> > >
> > > To clarify this point, we have revised the corresponding section of the manuscript as follows.
> > > We now introduce a variance parameter in the sampling distribution: $x \sim \mathcal{N}(x^*, cI_d)$,
> > > where $c > 0$ is a small constant and $I_d$ is the identity matrix. The parameter $c$ controls the concentration of the distribution around $x^\ast$: smaller values of $c$ produce points more tightly clustered near the geometric median, ensuring that the roots of the sampled tree systems lie close to $x^\ast$. The constant $c$ is treated as a tuning parameter in our experiments. Since this work represents our first implementation of the proposed FW-TSW method, we explored different values of $c$, but observed that the data is typically normalized and that setting $c=1$ consistently yields stable and competitive performance across datasets. To avoid potential confusion and improve readability, we therefore decided not to explicitly include $c$ in the main presentation of the method.
> > >
> > > **Q3. Choice of Central Point: Why was the geometric median chosen over simply using the barycenter of the measures in the tree construction? Is it because of Wasserstein-1 instead of Wasserstein-2? Did you try using the barycenter?**
> > >
> > > **Answer Q3.** The goal of our proposed sampling method is to choose the root of each tree slice near the Fermat–Weber point of the underlying data distribution, that is, the point that minimizes the expected distance to the data (Equation (10)):
> > >
> > > $x^* = \text{argmin}\_{x \in \mathbb{R}^d} \int_{\mathbb{R}^d} \|x - y\|_2 \, d\lambda(y).$
> > >
> > > However, in practice the true distribution is unknown and we only observe samples drawn from it. Therefore, we approximate the Fermat–Weber point using the geometric median computed from the available data points (Equation (11)):
> > >
> > > $x^* = \text{argmin}\_{x \in \mathbb{R}^d} \frac{1}{n} \sum\_{i=1}^n \|x - x_i\|_2.$
> > >
> > >
> > > This empirical geometric median provides a stable, data-dependent proxy for the ideal root location, allowing the sampled tree slices to better reflect the geometry of the observed distribution. Since Equation (11) is precisely the Monte Carlo approximation of (10), it is already in the barycentric “sense” mentioned by the Reviewer.
> > >
> > > **Q4. $k$-Disjoint Union: In Equation (6), what is the precise definition and calculation of the distance on the $k$-times disjoint union of $\mathbb{R}$ ?**
> > >
> > > **Answer Q4.** The tree system consists of $k$ lines whose union forms a connected subset of $\mathbb{R}^d$. The use of the disjoint union (followed by a quotient operation) on these $k$ lines serves to define the topology of the space obtained by gluing these lines together at a shared origin. The resulting structure is what we refer to as the tree system, which is treated as a tree-metric space in our work.
> > >
> > > The associated tree metric is defined as the length of the unique path joining any two points in this space. An intuitive depiction of the construction is shown in Figure 1 (Left), which the Reviewer may refer to for clarity.
> > >
> > > ---
> > > We sincerely thank the reviewer for their constructive feedback, which has helped improve our manuscript. We have updated the paper to reflect these changes and hope our responses effectively address the concerns. We look forward to any further engagement during the discussion phase.

---

### Official Review · Reviewer_cvC1 · 2025-11-01

**Soundness:** 3
**Presentation:** 3
**Contribution:** 2
**Rating:** 6
**Confidence:** 4

**Summary:**

This paper proposes the Fermat-Weber Tree-Sliced Wasserstein (FW-TSW) to solve a geometric inconsistency in the standard tree sliced Wasserstein (TSW) framework. While TSW is an $L_1$-based metric ($W_1$) and its tree-sampling is position-dependent, existing methods sample this position (the tree root $x$) from a Gaussian centered at the $L_2$ data mean. FW-TSW corrects this mismatch by centering its sampling distribution $\sigma_{FW}$ at the $L_1$ geometric median ($x^*$) which is the solution to the Fermat-Weber problem. This scheme is claimed to better capture data geometry and improve performance, adding only a negligible upfront cost to compute the median.

**Strengths:**

- This paper’s proposal to center the sampling distribution at the $L_1$ geometric median (the Fermat-Weber point) is the principled solution, aligning the geometry of the sampling space with the geometry of the metric.

- Experiment results, particularly in generative modeling (Table 3), are significant. The FW-TSW* variant achieves FID score (2.315) by making both the position ($x^*$) and directions ($\theta_i$) dependent on the measures being compared. This provides strong evidence that for high-dimensional generative tasks, data-agnostic slicing (like uniform sampling) is suboptimal, and a data-dependent discrepancy is superior.

- The cost of solving for the geometric median ($O(Tnd)$) is paid once per batch and does not dominate $O(Lkn \log n + Lkdn)$ TSW computation. The paper shows that this significant performance gain (e.g., in Table 3) is achieved with a negligible increase in wall-clock time, making it a drop-in replacement.

**Weaknesses:**

- FW-TSW is no longer a metric.

- The paper introduces two ideas: (1) data-dependent *positional* sampling (the core FW insight) and (2) data-dependent *directional* sampling (the FW-TSW* variant, eq 17). The paper’s narrative is built around the principled Fermat-Weber contribution, but the best empirical result (FID in Table 3) relies on the FW-TSW* variant, which includes the more heuristic directional sampling. This makes it difficult to attribute the performance. It is not clear if the central insight (the geometric median) is the key driver, or if the (less-justified) directional sampling is doing most of the work.

**Questions:**

- In the 25 Gaussians experiment (Table 1), TSW and FW-TSW methods achieve a better final error compared to SWGG, which converges faster initially. What properties of tree-slicing give this better final convergence, and does the Fermat-Weber centering improve the global optimization landscape beyond the standard TSW's mean-centering?

- The geometric median computation adds an $O(Tnd)$ cost (with $T = 100$) per batch, compared to the baseline's $O(nd)$ mean calculation. Is this iterative cost truly negligible for large $n$ and $d$?

---

> ### Author Response · Authors · 2025-11-27
> **Response [1/2]**
>
> We are grateful for the Reviewer’s feedback. To ensure a structured response, we have addressed related questions and concerns together. Please find the revised manuscript with all modifications highlighted in blue.
>
> ---
>
> **W1. FW-TSW is no longer a metric.**
>
> **Answer W1.** Both proposed metrics, although only semi-metrics, satisfy non-negativity, symmetry, and the identity of indiscernibles. While non-negativity and symmetry are standard and can be verified directly, we emphasize that the identity of indiscernibles is the most crucial property. It ensures that if the distance between two distributions is zero, then the two distributions must be identical. This guarantees that during training, where one typically minimizes the distance between the model-induced distribution and the empirical distribution, a zero loss indeed corresponds to the model distribution matching the empirical one.
>
> We also note that in many works introducing metrics between distributions, the triangle inequality fails to hold as well (see, for example, [1, 2]).
>
>
> **W2. The paper introduces two ideas: (1) data-dependent positional sampling (the core FW insight) and (2) data-dependent directional sampling (the FW-TSW$^{\ast}$ variant, eq 17). The paper’s narrative is built around the principled Fermat-Weber contribution, but the best empirical result (FID in Table 3) relies on the FW-TSW$^{\ast}$ variant, which includes the more heuristic directional sampling. This makes it difficult to attribute the performance. It is not clear if the central insight (the geometric median) is the key driver, or if the (less-justified) directional sampling is doing most of the work.**
>
> **Answer W2.** We thank the Reviewer for highlighting the two main components of our contributions, which correspond to the two proposed methods:
>
> FW-TSW: data-dependent positional sampling
>
> FW-TSW$^{\ast}$: data-dependent positional and directional sampling
>
> We emphasize that directional sampling is merely an additional component built on top of the core FW insight; FW-TSW* incorporates **both positional and directional sampling**, not only the latter.
>
> Our experimental results also support this interpretation. Both FW-TSW and FW-TSW$^{\ast}$ outperform the existing baselines, and FW-TSW$^{\ast}$ improves over FW-TSW only by a small margin. For instance, in our largest experiment, the Diffusion task, the best and second-best baseline scores are $2.53$ and $2.60$, whereas FW-TSW achieves $2.336$ and FW-TSW* further improves this only slightly to $2.315$.
>
> This confirms our claim: positional sampling provides the dominant contribution, while directional sampling acts as a modest enhancement rather than a core driver of performance.

---

> > ### Author Response · Authors · 2025-11-27
> > **Response [2/2]**
> >
> > **Q1. In the 25 Gaussians experiment (Table 1), TSW and FW-TSW methods achieve a better final error compared to SWGG, which converges faster initially. What properties of tree-slicing give this better final convergence, and does the Fermat-Weber centering improve the global optimization landscape beyond the standard TSW's mean-centering?**
> >
> > **Answer Q1.** We thank the reviewer for raising this question. Indeed, this is a primary motivation for using FW-TSW for improved sampling of the tree. Similar to Sliced Wasserstein (SW) distance, where prior works have proposed improving the sampling of slicing directions [1, 3] to demonstrate significant improvements, we aim to achieve similar gains in the TSW setting. In this work, we propose a new sampling scheme for tree slices that incorporates a geometric median-based sampling of the tree root. Within the context of TSW, this constitutes a novel idea. Moreover, unlike in SW variants, there is no direct analogue of this approach: the presence of a root in tree-metric systems is unique to TSW, encoding the positional structure of the slice, whereas SW relies solely on directional information. Our experiments confirm the benefit of this improved sampling in FW-TSW compared to the random sampling in Db-TSW.
> >
> > **Q2. The geometric median computation adds an $O(Tnd)$ cost (with $T=100$) per batch, compared to the baseline's $O(nd)$ mean calculation. Is this iterative cost truly negligible for large $n$ and $d$**
> >
> > **Answer Q2.** To accurately quantify the computational overhead of the geometric median, we conducted a controlled comparison between Db-TSW (utilizing the mean, $\mathcal{O}(nd)$) and FW-TSW (utilizing the geometric median, $\mathcal{O}(Tnd)$). We used a high-dimensional setting ($d=3000$) mirrors our Diffusion Model experiments, varying the batch size $n \in [100, 500]$ and fixing the Weiszfeld iteration count at $T=100$ to represent a worst-case scenario.
> >
> > As illustrated in the table below and Figure 4 (Appendix D.1), the overhead results in an approximate 11% increase in runtime at $n=500$ (89.46 ms vs. 80.57 ms).  Despite this increase in isolated computation, we maintain that the cost is negligible in practical applications. While the reported runtime assumes a fixed $T=100$, we observe that the algorithm typically converges in approximately 10 iterations. Furthermore, in end-to-end tasks such as training Diffusion Models, this difference is effectively absorbed by the substantial computational costs of backpropagation and forward passes; empirically, we observed that both Db-TSW and FW-TSW require approximately 85 seconds per epoch, making the cost of the geometric median insignificant in the overall training pipeline.
> >
> >
> > | $N$ | Db-TSW Runtime (ms) | FW-TSW Runtime (ms) |
> > | :---: | :---: | :---: |
> > | 100 | $13.90 \pm 0.36$ | $23.29 \pm 4.38$ |
> > | 200 | $27.94 \pm 0.56$ | $39.10 \pm 4.30$ |
> > | 300 | $48.04 \pm 1.98$ | $56.70 \pm 2.95$ |
> > | 400 | $64.04 \pm 1.67$ | $72.59 \pm 2.75$ |
> > | 500 | $80.57 \pm 0.66$ | $89.46 \pm 2.45$ |
> >
> > ---
> > We sincerely thank the reviewer for their constructive feedback, which has helped improve our manuscript. We have updated the paper to reflect these changes and hope our responses effectively address the concerns. We look forward to any further engagement during the discussion phase.
> >
> >
> > *Reference*
> >
> > [1] Nguyen et al., Sliced Wasserstein with Random-Path Projecting Directions.
> >
> > [2] Eloi Tanguy et al., Sliced Optimal Transport Plans.
> >
> > [3] Deshpande et al. Max-Sliced Wasserstein Distance and its use for GANs.

---

### Official Review · Reviewer_7to6 · 2025-11-03

**Soundness:** 3
**Presentation:** 2
**Contribution:** 2
**Rating:** 6
**Confidence:** 2

**Summary:**

This paper proposes Fermat–Weber Tree-Sliced Wasserstein (FW-TSW) and FW-TSW* — new variants of the Tree-Sliced Wasserstein (TSW) distance that integrate positional information via the geometric median (Fermat–Weber point).

The main idea: previous TSW approaches sample tree intersection points (roots) heuristically (typically from a Gaussian centered at the data mean). In contrast, FW-TSW replaces this with a data-adaptive distribution centered at the geometric median of the combined source–target supports, computed via Weiszfeld’s algorithm.

This modification yields:

Better alignment between positional sampling and data geometry;

Theoretical guarantees: semi-metricity, Euclidean invariance, and boundedness;

Empirical gains across gradient flows, topic modeling, and diffusion model training — with comparable runtime to TSW and Db-TSW.

**Strengths:**

### **Strengths**

- **Novel conceptual link:**
  Clever integration of *location theory* (Fermat–Weber / geometric median) into OT sampling, yielding a clean geometric interpretation that grounds the stochastic tree construction in data geometry.

- **Sound theoretical guarantees:**
  The paper rigorously proves semi-metricity, symmetry, Euclidean invariance, and boundedness of the proposed FW-TSW distance. These properties ensure mathematical consistency while extending classical TSW theory.

- **Efficiency preserved:**
  Despite adding geometric-median computation via Weiszfeld’s algorithm, the complexity remains $O(Lkn\log n + Lkdn)$, with only minor $O(Tnd)$ overhead — maintaining the practical efficiency of TSW.

- **Strong empirical performance:**
  FW-TSW and FW-TSW consistently outperform SW, TSW, and Db-TSW baselines on gradient flow, topic modeling, and diffusion model tasks. Results show improved convergence, stability, and lower FID without sacrificing runtime.

**Weaknesses:**

### **Weaknesses**

- **Limited generality of the tree structure.**
  The proposed FW-TSW and FW-TSW* are defined only on *star-shaped* tree systems, where all branches share a single root point (Eq. 15 / 18).
  More general tree geometries—e.g., unions of multiple disjoint lines or hierarchical branching—cannot be represented within this framework.
  This limits the flexibility of the model and its ability to capture more complex spatial relationships.

- **Unclear computational overhead.**
  The complexity of Weiszfeld’s algorithm (used to compute the geometric median) is not analyzed or included in the overall runtime discussion.
  While each iteration is \(O(nd)\), the number of iterations and convergence behavior can vary depending on data geometry, which may affect scalability in high dimensions.

- **Loss of metric property.**
  FW-TSW and FW-TSW* do not satisfy the triangle inequality because the sampling distribution \(\sigma_{\text{FW},\mu,\nu}\) depends on the data pair \((\mu,\nu)\).
  As discussed around Eq. (20), the distance is only a *semi-metric* when the sampling distribution is fixed.
  This data-dependent design improves adaptivity but sacrifices the strict metric property of the original TSW.

- **Minor issue:**
  Line 349 contains a small typo: “Equation equation 15.”

**Questions:**

1. Could you clarify whether the FW-TSW framework can be extended beyond the star-shaped tree assumption?
   In particular, is it possible to construct tree systems with multiple intersection points or hierarchical branches under the same theoretical formulation?

2. How many iterations are typically required for the Weiszfeld algorithm in your experiments?
   Did you observe any stability or convergence issues for high-dimensional datasets?

3. In particular, what is the benefit of \sigma_{dir,mu,\nu}  compared with U(S^{d-1})?

---

> ### Author Response · Authors · 2025-11-27
> **Response [1/2]**
>
> We are grateful for the Reviewer’s feedback. To ensure a structured response, we have addressed related questions and concerns together. Please find the revised manuscript with all modifications highlighted in blue.
>
> ---
> **W1. Limited generality of the tree structure.
> The proposed FW-TSW and FW-TSW$^\ast$ are defined only on star-shaped tree systems, where all branches share a single root point (Eq. 15 / 18). More general tree geometries—e.g., unions of multiple disjoint lines or hierarchical branching—cannot be represented within this framework. This limits the flexibility of the model and its ability to capture more complex spatial relationships.**
>
> **Q1. Could you clarify whether the FW-TSW framework can be extended beyond the star-shaped tree assumption?
> In particular, is it possible to construct tree systems with multiple intersection points or hierarchical branches under the same theoretical formulation?**
>
> **Answer W1+Q1.** We thank the Reviewer for the comment. While we acknowledge that restricting to a star-shaped tree may reduce the ability to capture more complex geometric structures compared to more general tree topologies, we choose to retain this setting due to its computational efficiency, which is comparable to that of the original Sliced Wasserstein distance. This design choice is also consistent with prior work on tree-sliced Wasserstein methods [1, 2], where the same star-shaped construction is adopted.
>
> The FW-TSW framework can, in principle, be extended beyond the star-shaped tree structure, as the Weiszfeld algorithm used for sampling only determines the root of the tree. However, for a general tree structure as formulated in [3], this sampling procedure affects only the choice of the root, while many branches of the tree do not originate near this point (which approximates the geometric median). As a result, several projection directions may lie far from the desired central region, undermining our goal of keeping all rays concentrated around the geometric median.
>
> More importantly, as previously discussed, adopting a general tree structure significantly increases computational cost, making it far less efficient than the star-shaped design.
>
> **W2. Unclear computational overhead.
> The complexity of Weiszfeld’s algorithm (used to compute the geometric median) is not analyzed or included in the overall runtime discussion.
> While each iteration is (O(nd)), the number of iterations and convergence behavior can vary depending on data geometry, which may affect scalability in high dimensions.**
>
> **Q2. How many iterations are typically required for the Weiszfeld algorithm in your experiments?
> Did you observe any stability or convergence issues for high-dimensional datasets?**
>
> **Answer W2 + Q2.** We clarify that the complexity of Weiszfeld’s algorithm and our overall framework are $\mathcal{O}(Tnd)$ and $\mathcal{O}(Lkn\log n + Lkdn + Tnd)$, respectively. This analysis is provided in Section 4.2 ("Computational Complexity") and detailed further in Appendix D.1. In all of our experiments and runtime comparisons, we set the maximum number of iterations $T=100$.
>
> To address the reviewer's concern regarding convergence behavior in high-dimensional or complex settings, we conducted additional experiments measuring the number of iterations $T$ required to reach convergence (tolerance $\epsilon=10^{-6}$, $n=100$). We evaluated this on varying data dimensions and different data geometries.
>
> As shown in the table below, Weiszfeld’s algorithm remains stable. Notably, convergence behavior remains robust in high-dimensional settings. The algorithm converges in roughly 10 iterations for the real-world CIFAR-10 dataset (where each sample is a flattened image, $d=3072$) and 6 iterations fo Gaussians $d=500$. While more complex data geometries (e.g., 25-Gaussians) require more iterations, the number of iterations remains low (maximum 74 iterations).
>
> | Dataset | Dimensions ($d$) | Iterations (Mean $\pm$ Std) | Max Iterations |
> | :--- | :---: | :---: | :---: |
> | *High-Dimensional Data* | | | |
> | CIFAR-10 | 3072 | $10.3 \pm 0.8$ | 11 |
> | Gaussian 500D | 500 | $5.6 \pm 0.5$ | 6 |
> | *Complex Geometry* | | | |
> | Gaussian 2D | 2 | $31.1 \pm 4.6$ | 39 |
> | 25-Gaussians | 2 | $29.8 \pm 19.0$ | 74 |
> | 8-Gaussians | 2 | $18.1 \pm 1.7$ | 22 |
> | Swiss Roll | 2 | $19.3 \pm 1.2$ | 21 |

---

> > ### Author Response · Authors · 2025-11-27
> > **Response [2/2]**
> >
> > **W3. Loss of metric property.
> > FW-TSW and FW-TSW$^{\ast}$ do not satisfy the triangle inequality because the sampling distribution $(\sigma_{\text{FW},\mu,\nu})$ depends on the data pair $((\mu,\nu))$.
> > As discussed around Eq. (20), the distance is only a semi-metric when the sampling distribution is fixed.
> > This data-dependent design improves adaptivity but sacrifices the strict metric property of the original TSW.**
> >
> >
> > **Answer W3.** Both metrics, although only semi-metrics, satisfy non-negativity, symmetry, and the identity of indiscernibles. While non-negativity and symmetry are standard and can be verified directly, we emphasize that the identity of indiscernibles is the most crucial property. It ensures that if the distance between two distributions is zero, then the two distributions must be identical. This guarantees that during training, where one typically minimizes the distance between the model-induced distribution and the empirical distribution, a zero loss indeed corresponds to the model distribution matching the empirical one.
> >
> > We also note that in many works introducing metrics between distributions, the triangle inequality fails to hold as well (see, for example, [4, 5]).
> >
> >
> > **W4. Minor issue:
> > Line 349 contains a small typo: “Equation equation 15.”**
> >
> > **Answer W4.** We thank the Reviewer for pointing this out. We have incorporated the necessary changes in the revised version.
> >
> > **Q3. In particular, what is the benefit of $\sigma_{dir,\mu,\nu}$ compared with $U(S^{d-1})$?**
> >
> > **Answer Q3.** The sliced-sampling distribution $\sigma_{dir,\mu,\nu}$ encourages the lines in the tree system to align with directions that connect data points to points sampled from the model distribution. This alignment can potentially accelerate the training process by focusing projections on informative directions. Similar ideas have also been adopted in prior works on slice-sampling strategies for sliced optimal transport; see, for example, [4].
> >
> > ---
> > We sincerely thank the reviewer for their constructive feedback, which has helped improve our manuscript. We have updated the paper to reflect these changes and hope our responses effectively address the concerns. We look forward to any further engagement during the discussion phase.
> >
> > *Reference*
> >
> > [1] Tran et al., Distance-Based Tree-Sliced Wasserstein Distance.
> >
> > [2] Thanh et al., Tree-Sliced Wasserstein Distance with Nonlinear Projection.
> >
> > [3] Tran et al., Tree-Sliced Wasserstein Distance: A Geometric Perspective.
> >
> > [4] Nguyen et al., Sliced Wasserstein with Random-Path Projecting Directions.
> >
> > [5] Eloi Tanguy et al., Sliced Optimal Transport Plans.

---

### Author Response · Authors · 2025-12-03
**General Response**

Dear Area Chair and Reviewers,

We sincerely appreciate the Area Chair's significant work to manage our submission after the unexpected events. Your support is greatly valued.

We also thank the reviewers for their thoughtful comments, which have helped us significantly improve the manuscript. We are pleased to note that reviewers generally recognize our strengths:
- Reviewers highlighted the importance of investigating sampling strategies for Tree-Sliced Wasserstein (TSW) [jHMb], recognizing our integration of the Fermat-Weber problem into the sampling process as novel [7to6] and principled [cvC1].
- Reviewers acknowledged the sound theoretical grounding, noting the rigorous proofs of semi-metricity, symmetry, Euclidean invariance, and boundedness [7to6].
- Reviewers highlighted the strong empirical results across diverse applications and that the method maintains the practical efficiency of TSW [7to6, cvC1, jHMb].

We are thankful for the suggestions from Reviewer, which we have addressed in the revised paper with all modifications highlighted in blue. The main concerns from Reviewer are as follow:

**FW-TSW is a semi-metric.** Reviewers 7to6 and cvC1 noted that FW-TSW does not satisfy the triangle inequality. We clarified that while FW-TSW is a semi-metric, it retains non-negativity, symmetry, and, most importantly, the identity of indiscernibles. This ensures that a zero FW-TSW value guarantees the two distributions are identical, making it a valid objective for model training.

**Computational cost of Weiszfeld algorithm.** Reviewers 7to6 and cvC1 questioned whether the Weiszfeld algorithm would introduce significant computational overhead. We referred to Appendix D.1 for a runtime analysis, demonstrating that the algorithm converges rapidly and adds negligible overhead. In real-world applications (e.g., Diffusion Models), the training wall-clock time of FW-TSW is identical to that of standard TSW.

**Limited Impact.** Reviewer jHMb raised concerns about the impact of the work as TSW is not yet popularly used. We emphasized that that improving sampling strategies remains an active line of research for Sliced-Wasserstein (SW). Our work extends this effort to TSW by exploiting its unique positional structure, which has no direct analogue in SW. Empirically, we demonstrate TSW's strong potential: in our largest Diffusion Model experiment, the best SW variant achieves an FID of 2.70, while FW-TSW achieves 2.336. These results underscore the value of exploring sampling strategies for TSW.

**Bound Tightness.** Reviewer jHMb questioned the tightness of our theoretical bound due to an additive constant. We clarified that this constant decays super-exponentially with respect to dimension $d$. Consequently, the bound becomes increasingly tight and informative in high-dimensional settings. We further noted that this constant remains small in practice.

Reviewer jHMb also dedicatedly suggested improvements to our writing and notations, which we have incorporated into the paper.

As the discussion period concluded earlier than anticipated, we were unable to receive further input from the reviewers. Consequently, we kindly request that the AC review our paper, rebuttal, and additional results, with regard to how each raised concern has been addressed. We believe that, had the discussion continued, the Reviewers would have been satisfied with our responses and would have considered updating their scores.

Warm regards,

The Authors

---

### Meta-Review · Area_Chair_iGSn · 2026-01-05

**Summary:**

The novelty and quality of being a principled solution were recognised by reviewers.

**Reviewer Concerns:**

Computational overhead has been made clear, but the proposed FW-TSW is indeed not a metric.

**Reviewer Scores:**

Questions raised by Reviewer jHMb have been addressed by authors but not updates from this reviewer.

---

### Decision · Program_Chairs · 2026-01-26

Accept (Poster)